# Modeling cardiac fibroblast heterogeneity from human pluripotent stem cell-derived epicardial cells

Ian Fernandes[1,2,3,10], Shunsuke Funakoshi [1,9,10] ✉, Homaira Hamidzada [4,5,6], Slava Epelman [4,5,6,7,8] & Gordon Keller [1,2,3] ✉

Cardiac fibroblasts play an essential role in the development of the heart and are implicated in disease progression in the context of fibrosis and regeneration. Here, we establish a simple organoid culture platform using human pluripotent stem cell-derived epicardial cells and ventricular cardiomyocytes to study the development, maturation, and heterogeneity of cardiac fibroblasts under normal conditions and following treatment with pathological stimuli. We demonstrate that this system models the early interactions between epicardial cells and cardiomyocytes to generate a population of fibroblasts that recapitulates many aspects of fibroblast behavior in vivo, including changes associated with maturation and in response to pathological stimuli associated with cardiac injury. Using single cell transcriptomics, we show that the hPSC-derived organoid fibroblast population displays a high degree of heterogeneity that approximates the heterogeneity of populations in both the normal and diseased human heart. Additionally, we identify a unique subpopulation of fibroblasts possessing reparative features previously characterized in the hearts of model organisms. Taken together, our system recapitulates many aspects of human cardiac fibroblast specification, development, and maturation, providing a platform to investigate the role of these cells in human cardiovascular development and disease.

Cardiac fibroblasts make up approximately 15–30% of the total cells in the heart and have been shown to play important roles in the development of the fetal heart and in maintaining homeostasis in the adult organ[1]. The majority of the cardiac fibroblast population derives from the epicardium, a protective mesothelial layer that surrounds the heart early in development and persists throughout adult life[2]. The epicardium develops from progenitors known as pro-epicardial cells that are specified from a structure positioned near the posterior region of the inferior pole of the developing heart known as the septum transversum mesenchyme[3]. Fibroblasts are generated from the epicardium through an epithelial-to-mesenchymal transition (EMT) that is mediated in part by retinoic acid (RA) secreted from the epicardium and TGFβ produced by the adjacent cardiomyocytes[4,5]. As they are specified, the newly formed fibroblasts migrate to and populate the

[1]McEwen Stem Cell Institute, University Health Network, Toronto, ON M5G1L7, Canada. [2]Department of Medical Biophysics, University of Toronto, Toronto, ON M5G1L7, Canada. [3]Princess Margaret Cancer Center, University Health Network, Toronto, ON M5G1L7, Canada. [4]Toronto General Hospital Research Institute, University Health Network Toronto, Toronto, ON M5G1L7, Canada. [5]Ted Rogers Centre for Heart Research, Translational Biology and Engineering Program, Toronto, ON M5G1L7, Canada. [6]Department of Immunology, University of Toronto, Toronto, ON M5G1L7, Canada. [7]Department of Laboratory Medicine and Pathobiology, University of Toronto, Toronto, ON M5G1L7, Canada. [8]Peter Munk Cardiac Centre, University Health Networr, Toronto, ON M5G1L7, Canada. [9]Present address: Center for iPS Cell Research and Application, Kyoto University, Kyoto 606-8507, Japan. [10]These authors contributed equally: Ian Fernandes, Shunsuke Funakoshi. ✉e-mail: s.funakoshi@cira.kyoto-u.ac.jp; Gordon.Keller@uhnresearch.ca

developing ventricular and atrial myocardium. The cardiac epicardial cells and derivative fibroblasts are distinguished from mesothelial cells and fibroblasts of other organs by the expression of a cohort of genes typically associated with the cardiomyocyte lineage including *GATA4*, *GATA6*, and *HAND2*[6].

One of the functions of cardiac fibroblasts is the secretion of extracellular matrix proteins that provide support for cardiomyocyte proliferation during fetal life, for the regulation of organ maturation in postnatal life and for structure in the adult heart. Analyses of the matrix at different stages have shown that the composition changes between fetal and adult life as the heart transitions from a developing tissue consisting of proliferative cells to a postnatal organ made up of quiescent cardiomyocytes that have exited cell cycle[7]. During fetal life, the matrix is comprised of a mixture of fibronectin, collagens, and various proteoglycans that functions to support the proliferation of fetal cells. Following birth, there is a shift to a collagen-rich matrix that is less supportive of proliferation and has a greater tensile strength to provide structure for the hemodynamics of the adult cardiac cycle[8]. Currently, it is not known if these changes in the matrix composition reflect the maturation of the population of fibroblasts or are mediated by other cells in the cardiac interstitium with the capacity to produce matrix.

In addition to their function in the normal heart, fibroblasts have been shown to play a key role in the response to tissue damage and diseases of the heart. In model organisms such as the zebrafish that possess cardiac regenerative capacity into adulthood, fibroblasts are activated following injury and deposit a unique fibronectin-rich ECM that is conducive to the proliferation of cardiomyocytes and revascularization of the newly formed myocardium[9,10]. Additionally, the activated cardiac fibroblasts are known to secrete soluble factors such as neuregulin1 (NRG1) and heparin-bound epidermal growth factor (HB-EGF) that promote cardiomyocyte proliferation and the anti-inflammatory molecule IL-10 that modulates inflammation in the damaged tissue[10,11]. As this regenerative phase progresses the activated response of the fibroblasts is downregulated by the secretion of Wntless (WLS), a WNT signaling modulator, from the cardiomyocytes[12]. This transient regulation of the fibrotic response is necessary to maintain the balance of injury resolution with cardiac regeneration[10].

In adult mammals with little cardiac regenerative capacity, ischemic insults such as myocardial infarction (MI) initiate a response in the fibroblast population that results in the formation of a non-contractile scar that salvages the structural integrity of the myocardium. Fibroblasts play an important role in mediating this repair process through the secretion of tissue-modulating factors and the remodeling of ECM components. Activation of fibroblasts appears to be a common response in other forms of cardiomyopathies as well as advanced stages of these diseases are associated with extensive fibrosis in the heart. Fibrosis can exist as interstitial or replacement fibrosis with the latter being associated with cardiomyocyte necrosis. The role of fibroblasts in these different diseases suggests that activation of the fibroblast population is a common response that persists in the absence of a regeneration program.

Although there is a large body of evidence regarding the dynamics of fibroblast activation and proliferation and scar formation in damaged hearts, the cues regulating these responses are poorly understood. Additionally, it is not known if all resident cardiac fibroblasts respond to injury or if the response is mediated by a specific subset of cells[13]. Recent single cell analyses of adult mouse cardiac fibroblasts have shown that the population is heterogeneous and contains subsets with potential reparative features. Farbehi et al. identified a subpopulation of fibroblasts based on expression of the WNT signaling antagonist WIF1 that appears to promote new blood vessel formation and participates in the regulation of the fibrotic response through the expression of WNT ligands and repressors[14]. In a more recent study, Villalba-Ruiz et al. described a subpopulation of

fibroblasts with similar reparative characteristics. These cells, identified by the expression of an enzyme involved in collagen processing, collagen triple helix repeat-containing protein 1 (CTHRC1), were found to be pro-angiogenic and essential for the scar-forming process[15]. The presence of these subpopulations of fibroblasts suggests that there may be a population, even in the adult context, that possesses the capacity for repair. Whether or not comparable populations exist in the human heart remains to be determined.

Our understanding of human cardiovascular development including the specification of specific cell types such as cardiac fibroblasts is limited due to the scarcity of human heart tissue and our ability to maintain and study it ex vivo. Given this, many investigators have turned to human pluripotent stem cells (hPSC) as a source of cardiovascular cells and as a model system to study human cardiovascular development and disease. Over the past decade, protocols have been developed that promote the generation of different cardiomyocyte subtypes as well as many non-cardiomyocyte lineages including epicardial cells from hPSCs[16–18]. In this study, we used the hPSC system to model the interactions of cardiomyocytes with epicardial cells in 3D organoid structures to recapitulate the events that lead to the development and emergence of the human cardiac fibroblast population. We show that these newly generated fibroblasts seed the organoid tissue, mature in these structures and display responses to pathological stimuli that are similar to those observed in fibroblasts in the failing heart. Single cell RNA sequencing analyses revealed a high degree of heterogeneity within the fibroblast population that recapitulates molecular features of the cardiac fibroblast population in the adult ischemic heart in vivo. These analyses also identified a subpopulation of human fibroblasts transcriptionally similar to the reparative cells described in the murine heart post-MI. Together, these findings demonstrate the power of using developmental biology approaches to generate hPSC-derived cardiovascular populations that display functional characteristics and disease responses comparable to the corresponding populations found in the adult heart.

## Results

### Generation of cardiac organoids using ventricular cardiomyocytes and epicardium

To model cardiac fibroblast development and function in vitro, we generated organoids consisting of hPSC-derived ventricular cardiomyocytes and epicardial cells. This approach was designed to mimic the known interactions between these two cell types in the early heart that result in the induction of EMT within the epicardial population and the subsequent production of derivative cell types including cardiac fibroblasts[19]. For such models to be accurate and predictable, however, it is important to use cell types that most closely correspond to those from the appropriate staged developing heart. We have previously reported on the generation of hPSC-derived ventricular cardiomyocytes and demonstrated that they represent an immature stage of development[17,20]. In an earlier study, we also described a protocol for the derivation of epicardial cells from hPSCs[16]. While this approach did promote the development of cells with epicardial characteristics, the levels of *ALDH1A2*, a key marker of this lineage, were variable within the population. Additionally, the mesoderm origin of the lineage was not investigated in that study.

To improve the generation of RALDH2[+] (*ALDH1A2*) epicardial lineage cells, we investigated the role of retinoic acid signaling at the mesoderm stage of development as studies in the mouse and chick suggest that the progenitors of the proepicardium are exposed to retinoid signaling in vivo[21]. Additionally, the mesoderm we used to generate epicardial cells expresses RALDH2 indicating that it represents a stage requiring active RA signaling (Supplementary Fig. 1A). For these analyses, RALDH2 expressing cells were measured and quantified using the Aldefluor assay[22]. Treatment of the developing population between day 4 and 6 with retinol (ROH), the substrate for RA

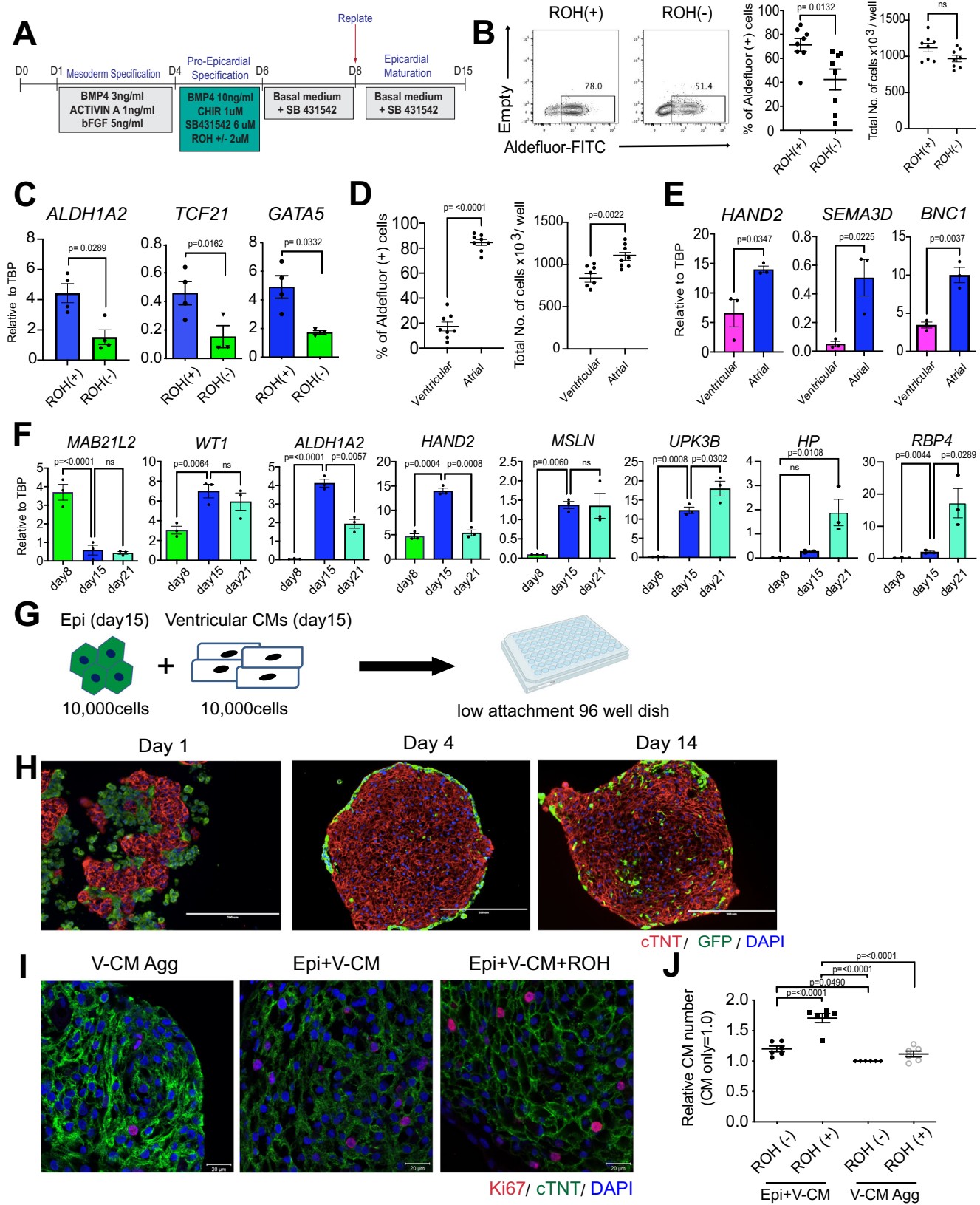

synthesis, did not alter the proportion of ALDH⁺ cells detected at day 8 of differentiation but significantly increased the proportion measured at day 15 (Fig. 1A, B, Supplementary Fig. 1B). The addition of ROH did not impact the total cell number, suggesting that RA functions to pattern the RALDH2⁺ lineage at the mesoderm stage of development. The addition of ROH also led to an increase in the expression levels of

*ALDH1A2, TCF21* and *GATA5* in the day 15 population (Fig. 1C). The expression levels of other genes associated with the epicardial lineage including *WT1, TBX18, GATA4, GATA6* and *HAND2* were not impacted (Supplementary Fig. 1C). These findings show that RA signaling at the mesoderm stage of development enhances the generation of a RALDH2⁺ epicardial population.

**Fig. 1 | Generation of cardiac organoids with ventricular cardiomyocytes and epicardial cells. A** Protocol to generate epicardial cells from human pluripotent stem cells. **B** Left; Representative flow cytometry analyses of Aldeflour (ALDH) in day15 retinol (ROH)-treated (ROH(+)) and untreated (ROH(-)) epicardial cells. Right; Quantification of the proportion of ALDH positive cells and total number of cells in ROH-treated and untreated epicardial population ($N = 8$ biologically independent samples). Statistical analysis was performed by two-sided unpaired $t$-test. **C** RT-qPCR expression analyses of *ALDH1A2*, *TCF21*, and *GATA5* in the indicated populations ($N = 4$ biologically independent samples). Statistical analysis was performed by two-sided unpaired $t$-test. **D** Quantification of the proportion of ALDH positive cells and total number of cells in ventricular and atrial mesoderm-derived epicardial cells ($N = 8$ biologically independent samples). Statistical analysis was performed by two-sided unpaired $t$-test. **E** RT-qPCR expression analyses of epicardial marker genes in the indicated populations ($N = 3$ biologically independent samples). Statistical analysis was performed by two-sided unpaired $t$-test. **F** RT-qPCR expression analyses of epicardial marker genes in the indicated time points

($N = 3$ biologically independent samples). Statistical analysis was performed by one-way ANOVA with Tukey's multiple comparisons. **G** Schema of the coculture of GFP-positive epicardial cells and ventricular cardiomyocyte derived from hPSCs. **H** Representative immunostaining of day 1, 4, and 14 cardiac organoids. Scale bar; 200 μm. ($N = 5$ biologically independent samples). **I** Representative immunostaining of Ki67 in ventricular cardiomyocyte aggregates, cardiac organoids, and cardiac organoids treated with ROH. Scale bar; 20 μm. **J** Quantification of the relative number of cardiomyocytes in the cardiac organoids and cardiomyocyte aggregates (V-CM Agg) treated as indicated ($N = 6$ biologically independent samples). The number of cardiomyocytes in V-CM only without ROH (ROH(-)) was defined as 1.0 and the relative number of cardiomyocytes was calculated in each condition. Statistical analysis was performed by one-way ANOVA with Tukey's multiple comparisons. All error bars represent SEM. All values shown for the PCR analyses are relative to the housekeeping gene TBP. V-CM ventricular cardiomyocytes. Epi epicardial cells. Source data are provided as a Source Data file. Created with BioRender.com.

The mesoderm used for the above studies has the same ALDH and CD235a/b expression profile and is induced under the same conditions (low concentrations of BMP4 and Activin A) as the mesoderm that gives rise to atrial cardiomyocytes[17] (Supplementary Fig. 1A). To determine if this mesoderm is the best source of these cells, we compared its epicardial potential to that of ALDH⁻/CD235a/b⁺ mesoderm induced with higher concentrations of these pathway agonists that generates ventricular cardiomyocytes (Supplementary Fig. 1D). As shown in Fig. 1D, the ALDH⁺/CD235a/b⁻ (atrial) mesoderm gave rise to a higher frequency and higher total number of cells than the ALDH⁻/CD235a/b⁺ (ventricular) mesoderm at day 15 of culture. The population derived from the ALDH⁺/CD235a/b⁻ (atrial) mesoderm also expressed significantly higher levels of genes associated with epicardial development including *HAND2*, *SEMA3D* and *BNC1* than the one generated from the ALDH⁻/CD235a/b⁺ (ventricular) mesoderm (Fig. 1E). Together, these findings provide strong evidence that the human epicardial lineage develops from ALDH⁺/CD235a/b⁻ 'atrial' mesoderm.

In the mouse embryo, the epicardial lineage develops from the pro-epicardial organ that is derived from a transient structure known as the septum transversum mesenchyme (STM) that can be identified by expression of the transcription factor *MAB21L2*[23]. The expression of *MAB21L2* is downregulated with the development of the proepicardial progenitors and the subsequent specification of the epicardial lineage. The establishment of the epicardium proper is defined by the expression of a cohort of genes including cardiac transcription factors (*WT1*, *HAND2*, *BNC1*, *GATA4/5*, *TCF21*) the secreted protein, *SEMA3D* and markers associated with an epithelial identity (*MSLN* and *UPK3B*)[23]. To stage the hPSC-derived lineage, we carried out a kinetic analysis and found that the STM marker, *MAB21L2* was expressed in the population on day 8 and then downregulated at days 15 and 21 (Fig. 1F). The levels of some of the epicardial genes including *WT1*, *ALDH1A2*, *HAND2*, *MSLN* and *UPK3B* were low at day 8, showed a significant upregulation by day 15 and then remained constant or declined by day 21. Other genes were expressed at similar levels at all 3 time points or upregulated in the final stages of culture including several adult epicardial markers such as *HP*, *RBP4*, *HAS1* and *NFATC1* (Fig. 1F, Supplementary Fig. 1E). Taken together, these patterns suggest that the transition from the pro-epicardial identity to the epicardial cells that wrap the heart occurs between days 8 and 15 with the transition to a more adult-like phenotype by day 21 of culture. Given that the levels of the fetal epicardial genes, *ALDH1A2* and *HAND2*, were highest on day 15, we chose to use this stage to co-culture with the ventricular cardiomyocytes (Fig. 1F).

To generate cardiomyocyte/epicardial organoids, $10^4$ cardiomyocytes and $10^4$ epicardial cells were mixed at a 1:1 ratio in 96 well round bottom plates (Fig. 1G). This format yielded single aggregates per well that were maintained in basal medium with no exogenous factors for two weeks. Immunohistochemistry analyses showed that the cells began to form aggregates within 24 h of culture. At this early

stage, there were no signs of organization within the structures. By day 4, however, there was obvious segregation of the cells, as the majority of the epicardial cells (GFP⁺) were now positioned at the periphery of the aggregates, surrounding the inner population of cardiomyocytes. Following an additional 10 days of culture, this segregation was partially lost as an increased number of GFP-positive cells were detected within the aggregates (Fig. 1H). Histological analysis showed that GFP-positive epicardial cells exhibited a rounded shape in the early-stage aggregates however, by day 14, they displayed an elongated morphology (Supplementary Fig. 1F).

To determine if this simple organoid model can recapitulate key interactions between the epicardium and myocardium in the embryonic heart, we investigated two different aspects of these interactions. The first was the promotion of cardiomyocyte proliferation by the epicardial cells and the second was the induction of EMT in the epicardial cells following contact with cardiomyocytes. Comparison of organoids to cardiomyocyte aggregates (no epicardial cells, hereafter referred to as ventricular cardiomyocyte aggregates, VCM-aggs) revealed that the former had a higher percentage of Ki67-positive cardiomyocytes and a greater number of total cardiomyocytes than the latter (Fig. 1I, J, Supplementary Fig. 1G) indicating that the presence of epicardial cells does, indeed increase cardiomyocyte proliferation. The addition of ROH to the cultures led to a further increase in the total number of cardiomyocytes in the organoids but not in VCM-aggs (Fig. 1J). The addition of ROH however, did not significantly affect the expression level of immature sarcomere-related (*MYH6*), Ca²⁺ handling-related (*ATP2A2*), and ion channel-related (*HCN2*, *HCN4*, *CACNA1C*, *KCNK1*) genes in the cardiomyocytes, suggesting that this stimulus with or without the organoid environment does not promote the electrophysiological maturation of VCMs (Supplementary Fig. 1H).

## Epicardial-cardiomyocyte interactions in the organoids

As a measure of the initiation of EMT and the development of epicardial-derived cell types (EPDCs), we analyzed the organoids at different time points by flow cytometry for the presence of CD90⁺ mesenchymal cells and ALDH⁺ epicardial cells. For these studies, cardiomyocytes were generated from HES2 hPSCs engineered to constitutively express RFP and the epicardial cells from HES2 WT hPSCs. This strategy enabled us to specifically analyze the progeny from the RFP⁻ epicardial cells (Fig. 2A). Prior to culture, approximately half of the epicardial cells were ALDH⁺ and of these, 30% expressed low levels of CD90 (CD90ˡᵒʷ) (Fig. 2B). A low frequency of CD90⁺ALDH⁻ cells was also detected. By 4 days of culture, the patterns changed as 50% of the cells were CD90⁺ALDH⁺ and more than 20% were CD90⁺ALDH⁻. The increase in the proportion of CD90⁺ cells suggests that EMT has already initiated. At 14 days of culture, the organoids contained 3 predominant populations: CD90ʰⁱᵍʰALDH⁻/ˡᵒʷ cells (85%), CD90⁻/

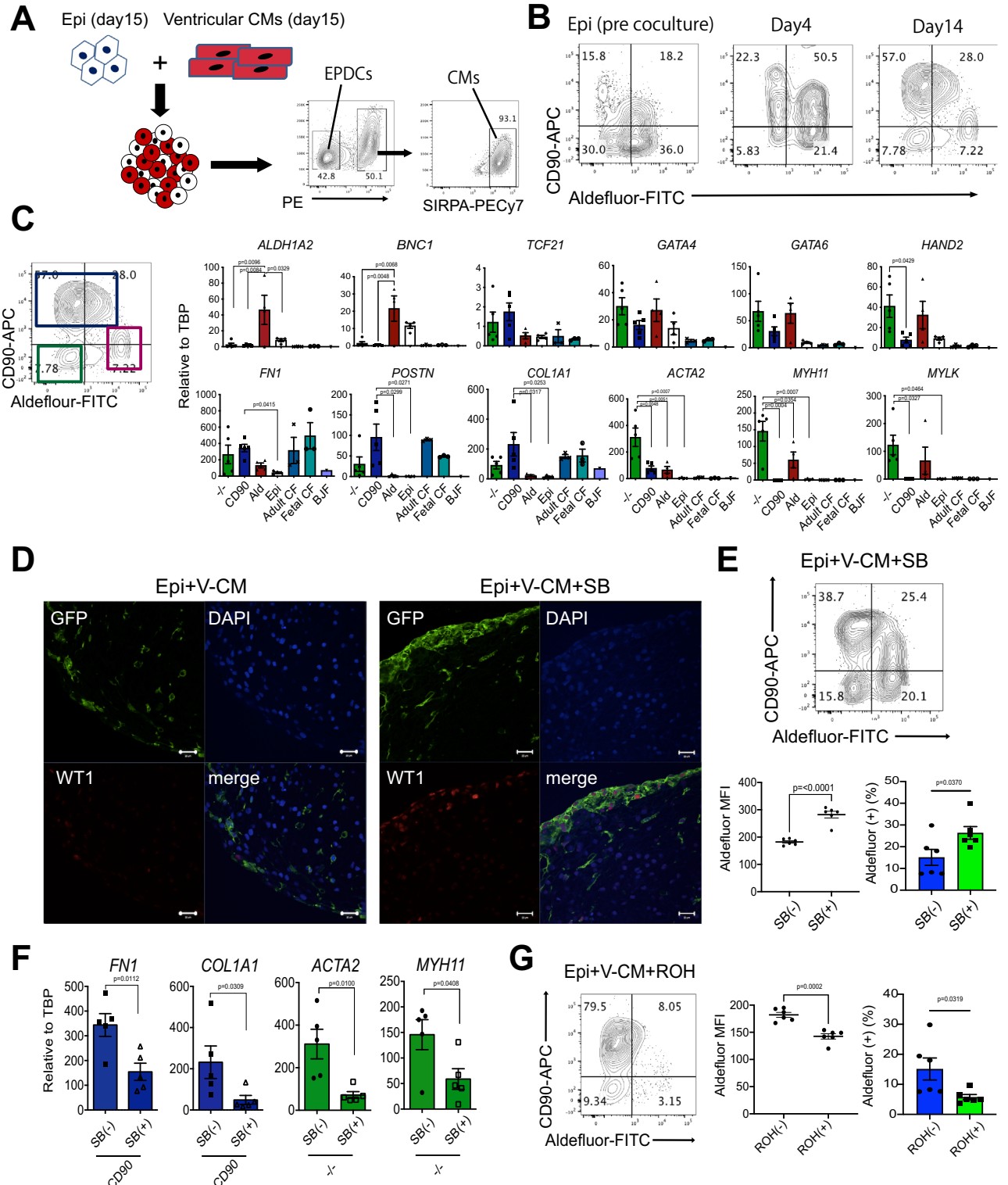

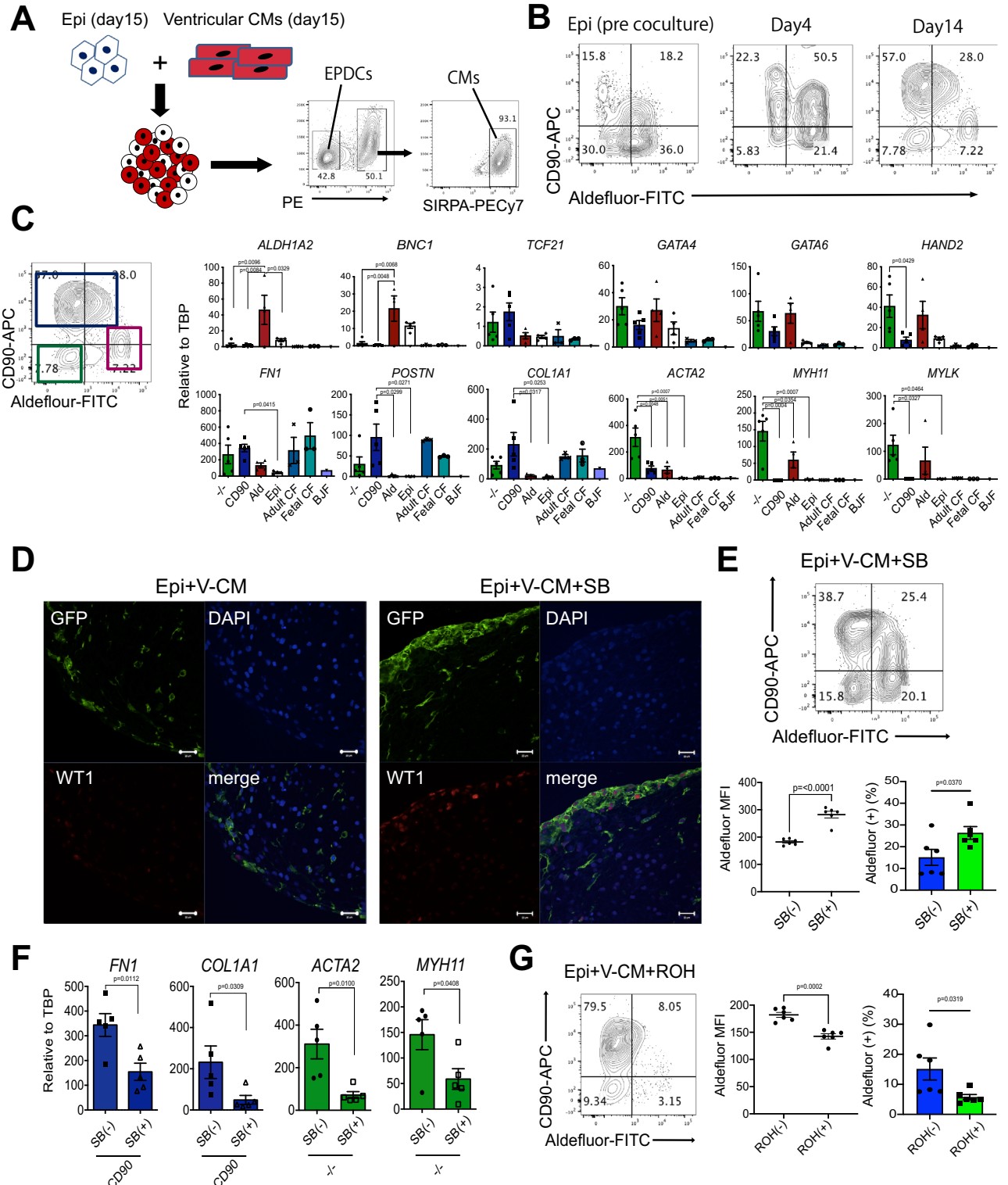

lowALDHhigh cells (7%) and CD90−ALDH−/low cells (8%). The large size of the CD90high population at this stage suggests that most of the epicardial cells had undergone EMT and generated mesenchymal progeny. The upregulation of expression of *SNAI1* and *TWIST1* in the CD90+ population supports the interpretation that the cells were generated through a EMT process (Supplementary Fig. 2A). To further characterize these populations, we isolated them by FACS and analyzed them for expression of genes indicative of fibroblast and smooth muscle cell development and for the persistence of epicardial cells. RT-PCR analyses showed that the CD90−/lowALDHhigh fraction (red square)

expressed epicardial-related genes (*ALDH1A2*, *BNC1*), the CD90highALDH−/low fraction (blue square) expressed genes associated with fibroblast development including *POSTN*, *FN1*, *COL1A1* whereas the CD90−ALDH−/low fraction (green square) displayed an expression profile indicative of the presence of coronary smooth muscle lineage cells (*ACTA2*, *MYH11*, and *MLYK*) (Fig. 2C). All three populations expressed cardiac transcription factors, including *GATA4*, *GATA6*, *HAND2*, and *TCF21*, indicating that the EPDCs in the organoids maintained cardiac-specific features (Fig. 2C). CD31+ endothelial cells were not detected in the populations throughout the culture period,

**Fig. 2 | Characterization epicardial-derived cells (EPDCs) in the cardiac organoids. A** Schema of the generation of the cardiac organoids using the non-labeled epicardial cells and RFP-positive ventricular cardiomyocytes derived from hPSCs, and the strategy to purify the EPDCs and the cardiomyocytes from the cardiac organoids by FACS. **B** Representative flow cytometry analyses of Aldefluor (ALDH) and CD90 expression in the cardiac organoids in the indicated time points. **C** Left; Representative flow cytometry analyses of 3 subpopulations in the organoids. Right; RT-qPCR expression analyses of epicardial (*ALDH1A2, BNC1*), transcription factor (*TCF21, GATA4, GATA6, HAND2*), extracellular matrix (*FN1, POSTN, COL1A1*), and smooth muscle (*ACTA2, MYH11, MYLK*) genes in the indicated populations (*N* = 5 biologically independent samples). Adult and fetal cardiac fibroblasts (CF) and skin BJ fibroblasts were included as a reference. Epi: day15 epicardium prior to coculture, CF: adult and fetal cardiac fibroblasts (*N* = biologically independent samples each). BJF: skin BJ fibroblasts (*N* = 1). **D** Representative immunostaining of WT1 and GFP (EPDCs) in the untreated and SB431542 (SB)-treated cardiac organoids. Scale bar; 20um. (*N* = 5 biologically independent samples). **E** Upper;

Representative flow cytometry analyses of ALDH and CD90 expression in the SB-treated cardiac organoids. Lower; Quantification of the proportion of ALDH+ cells in each population from the flow cytometry analyses (*N* = 6 biologically independent samples). **F** RT-qPCR expression analyses of *FN1* and *COL1* in CD90-positive populations in the SB-treated and untreated organoids (*N* = 5 biologically independent samples) and of *ACTA2* and *MYH11* in ALDH-negative / CD90-negative populations isolated (FACS) from the SB-treated and untreated organoids (*N* = 5 biologically independent samples). **G** Left; Representative flow cytometry analyses of ALDH and CD90 expression in the retinol (ROH)-treated cardiac organoids. Right; Quantification of the proportion of each population in the flow cytometry analyses (*N* = 6 biologically independent samples). Statistical analysis was performed by one-way ANOVA with Tukey's multiple comparisons in (**C**) and by two-sided unpaired *t*-test in (**E**)–(**G**). All error bars represent SEM. V-CM ventricular cardiomyocytes. Source data are provided as a Source Data file. Created with BioRender.com.

suggesting that our day 15 epicardial cells do not possess the potential to differentiate into these cells under these culture conditions (Supplementary Fig. 2B).

To determine if it was possible to generate fibroblasts with similar characteristics directly from the epicardial cells independent of organoid formation, we treated them with a combination of TGFβ1, EGF, bFGF and CHIR for 3 days then with bFGF alone for an additional 3 days (Supplementary Fig. 2C). Flow cytometric analyses showed that the majority of the cells within the population following this culture period were CD90+ and ALDH-, indicative of a transition to a fibroblast fate (Supplementary Fig. 2D). RT-qPCR analysis revealed that the cultured population downregulated the epicardial genes *WT1, TBX18, ALDH1A2* and *BNC1* and upregulated genes associated with fibroblast development including *FN1, POSTN* and *COL1A1* to levels similar to those observed in fetal and adult primary cardiac fibroblasts (Supplementary Fig. 2E). The transcription factors *GATA4* and *GATA6* were also expressed in the hPSC derived fibroblasts indicating that they displayed the molecular identity of cardiac fibroblasts (Fig. 2C, Supplementary Fig. 2E).

Immunostaining analyses showed that the organoids but not the VCM-aggs contained fibronectin (Supplementary Fig. 2F) indicative of the presence of functional fibroblasts in these structures. MYH11+ cells were also found in the organoids, confirming the finding from the RT-PCR analyses that cells with smooth muscle characteristics are generated (Supplementary Fig. 2G). Taken together, these observations show that, in the context of the organoids, the epicardial cells undergo EMT and give rise to progeny that display characteristics of cardiac fibroblasts and smooth muscle cells, recapitulating the developmental events observed in the embryonic heart.

Studies in mice have shown that TGFβ signaling plays a pivotal role in the EMT process and in the generation of EPDCs in the developing heart[24]. To determine if this pathway is also involved in these processes in the hPSC-derived organoids, we treated them with the TGFβ inhibitor, SB431542 (SB) and then analyzed the organoids for the presence of epicardial cells and EPDCs. Treatment with SB for 2 weeks resulted in the maintenance of a distinct GFP-positive WT1+ epicardial population surrounding the organoid (Fig. 2D, Supplementary Fig. 2H). Flow cytometric analyses showed that day 14 SB-treated organoids had a higher proportion of ALDH+ cells and a lower proportion of CD90+ cells than the untreated controls, confirming the presence of a larger epicardial population (Fig. 2E vs 2B). RT-PCR analyses of isolated populations revealed that expression levels of *FN1* and *COL1A1* were lower in the SB-treated than in the control CD90+ALDH- fibroblasts (Fig. 2F). Similarly, the SB-treated CD90-ALDH- cells expressed lower levels of *ACTA2* and *MYH11* than the untreated cells. In addition to the TGFB pathway, retinoid signaling also appears to promote EMT as organoids treated with ROH were found to contain significantly fewer ALDH+ cells than the untreated structures (Fig. 2G). RT-PCR analyses of

isolated populations showed that expression levels of *FN1, COL1A1, ACTA2,* and *MYH11* were not significantly different between treated and untreated populations, indicating that retinoid signaling has limited impact on the molecular characteristics of CD90+ fibroblast population (Supplementary Fig. 2I). Collectively, these observations indicate that EMT in the organoids is regulated in part by TGFβ and retinoid signaling (Supplementary Fig. 2J).

## Metabolic maturation of the cardiac organoids

For the organoids to be useful to model diseases that affect the adult heart, the cells should represent a stage of maturation similar to their counterparts in the postnatal organ. As the cells in the organoids and cardiomyocyte aggregates at this stage are immature, we next cultured them in the presence of a PPARα agonist (GW7475), T3 hormone, dexamethasone, and palmitate (PPDT) in low glucose-containing media for an additional 2 weeks, conditions we have previously shown promote metabolic maturation of hPSC-derived cardiomyocytes (Fig. 3A)[20]. Flow cytometric analyses of the organoids following this culture period showed that proportion of total RFP+ cells (~50%) and cTNT+ MLC2v+ cells (85–90%) within the RFP+ subpopulation were similar to that of the 2 week-old organoids (Fig. 2A, Supplementary Fig. 3A). The cardiomyocyte population in the treated organoids displayed expected characteristics of maturation observed in vivo including an increase in sarcomere length within the cells (Fig. 3B, Supplementary Fig. 3B), a reduction in the proportion of Ki67+ cells indicating an exit from the cell cycle (Fig. 3C) and an increase in CX43 protein expression (Fig. 3D). To functionally assess changes in the oxidative capacity of these populations, we used the Seahorse XF assay to measure the oxygen consumption rate (OCR) between immature and mature organoids and VCM-aggs. Cardiomyocytes cultured in both formats responded to the maturation stimuli as demonstrated by increases in basal respiration, spare capacity and maximal capacity. However, those cultured in the organoids showed higher levels of all these parameters, indicating that the presence of epicardial cells and derivative cell types enhances the metabolic changes in these cells (Fig. 3E).

Analyses of the treated organoids showed molecular changes within the cardiomyocyte population indicative of maturation, including upregulation of expression of sarcomere-related (*MYOZ2, MYOM3*), Ca2+ handling-related (*ATP2A2, HRC, CACNA1C*), mitochondrial (*CKMT2, COX7A1, MFN2*), fatty acid oxidation-related (*MLYCD, CD36, FABP3, CPT1B*) and ion channel-related (*KCNK1, HCN2, KCND3, CACNA1C, KCNJ2, KCNH2*) genes compared to the untreated, age-matched control organoids (Supplementary Fig. 3C). The expression levels of immature sarcomere genes (*MYH6* and *TNNI1*) and a pacemaker related ion channel (*HCN4*) were significantly downregulated. However, the expression of adult sarcomere genes (*MYH7* and *TNNI3*) was not significantly different than their immature counterparts

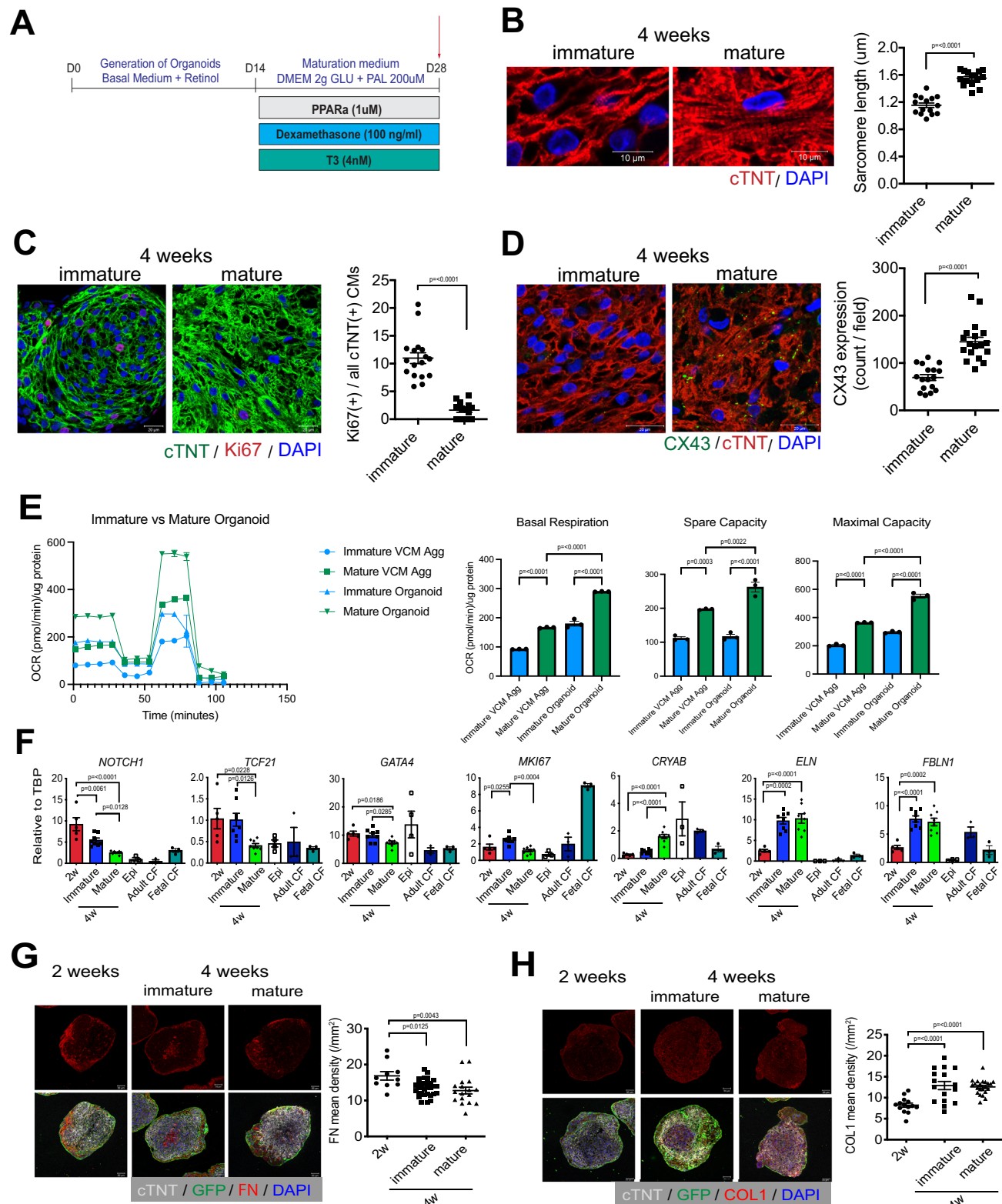

suggesting other cues may be missing from the culture to promote these changes.

Although the expression levels of genes associated with glycolysis (*GLUT1, GLUT4, PDK4*), were elevated in some experiments, the differences were not consistent and consequently not significant. Together, these findings show that the cardiomyocytes in the organoids underwent changes indicative of metabolic, electrophysiological, and

structural maturation, similar to the changes reported in our previous study[20].

To determine if the changes in culture conditions also promoted maturational changes in the epicardial-derived fibroblast populations, we isolated the CD90⁺ cells from the treated organoids and analyzed them for expression levels of genes that distinguish fetal and postnatal cardiac fibroblast populations[25,26]. This analysis revealed both time-

**Fig. 3 | Metabolic maturation of cardiac organoids. A** Protocol to induce metabolic maturation in the cardiac organoids. **B** Left; Representative immunostaining of cTNT in the immature and mature cardiac organoids. Scale bar; 10 μm. Right; Quantification of the sarcomere length in the indicated conditions ($N = 15$ from 3 independent biological replicates). **C** Left; Representative immunostaining of Ki67 and cTNT in the immature and mature cardiac organoids. Scale bar; 20 μm. Right; Quantification of the percentage of Ki67-positive cardiomyocytes in the organoid ($N = 17$ from 3 independent biological replicates). **D** Left; Representative immunostaining of CX43 and cTNT in the immature and mature cardiac organoids. Scale bar; 20 μm. Right; Quantification of the CX43 expression in the organoid ($N = 18$ from 3 independent biological replicates). **E** Analyses of the oxygen consumption rate (OCR) in the indicated populations using the Seahorse fatty acid oxidation (FAO) assay ($N = 3$ independent biological replicates each). **F** RT-qPCR expression analyses of indicated genes in the early (2 weeks) and late (4 weeks) CD90-positive cardiac fibroblast populations isolated (FACS) from immature and mature organoids ($N = 8$ biologically independent samples). Adult and fetal CFs were included as a reference ($N = 3$ biologically independent samples each). epi: day15 epicardial cells prior to the initiation of coculture ($N = 4$ biologically independent samples). **G** Left; Representative immunostaining of fibronectin (FN) in the cardiac organoids at the indicated time points. Scale bar; 20 μm. Right; Quantification of the FN density at the indicated time points ($N = 10$ for 2 W, $N = 30$ from 3 independent biological replicates). **H** Left; Representative immunostaining of collagen type 1 (COL1) in the cardiac organoids at the indicated time points. Scale bar; 20 μm. Right; Quantification of the COL1 density at the indicated time points ($N = 22$ from 3 independent biological replicates). Statistical analysis was performed by one-way ANOVA with Tukey's multiple comparisons in (**E**)−(**H**) and by two-sided unpaired t-test in (**B**)−(**D**). All error bars represent SEM. VCM Agg ventricular cardiomyocyte aggregates. CF cardiac fibroblasts. Source data are provided as a Source Data file.

dependent and factor-dependent maturation changes within these cells. Changes independent of the maturation cues included the upregulation of expression of *ELN* and *FBLN1*, genes that encode ECM proteins and the downregulation of *SOX4* and *HBEGF* genes associated with a fetal cardiac fibroblast identity (Fig. 3F, Supplementary Fig. 3D). These changes are consistent with the changes observed between the fetal and postnatal heart in vivo. The expression levels of other genes, such as *NOTCH1, TCF21, GATA4* and *MKI67*, were downregulated following culture in the maturation media, whereas *CRYAB* expression was upregulated (Fig. 3F). These differences are also reflective of changes in the fibroblasts in fetal and neonatal hearts. These observations show that the fibroblasts can respond to the same hormonal and metabolic cues governing cardiomyocyte maturation and undergo molecular changes associated with maturation of the population in the developing heart in vivo.

To determine if the changes in the fibroblast population were dependent on the environment of the organoid, we subjected the fibroblasts generated directly from epicardial cells to the maturation culture (Supplementary Fig. 3E). As these fibroblasts did not survive in the aggregate format, they were cultured and treated in 2D monolayers. RT-qPCR analysis following maturation culture showed that the changes in expression of *NOTCH1, TCF21, GATA4, MKI67* observed in the fibroblasts isolated from the organoid were not detected in the fibroblasts generated directly from the epi cells (Supplementary Fig. 3F). The expression of *CRYAB* was significantly upregulated as observed in the organoid fibroblasts and reflective of changes in the fibroblasts in fetal and neonatal hearts. These observations indicate that fibroblasts within the organoids more accurately mimic the in vivo maturation process than fibroblasts cultured on their own.

Maturation of the heart is associated with global changes in the ECM composition, characterized by a reduction in fibronectin and an increase in collagen proteins with the transition from fetal to postnatal life[27]. Immunohistochemistry analyses revealed similar changes in the organoids as the fibronectin content decreased between 2 and 4 weeks of culture, whereas the amount of collagen I increased during this timeframe (Fig. 3G, H). Culture in the maturation media did not impact these changes.

## Modeling cardiac injury in cardiac organoids

In our previous study, we showed that aggregates of mature cardiomyocytes (without fibroblasts) would undergo changes associated with cardiomyocyte injury following exposure to pathological stimuli including isoproterenol (ISO) and hypoxia[20]. As cardiomyocytes and cardiac fibroblasts are known to interact closely to facilitate remodeling during the progression to heart damage, we next asked if the injury response in the mature organoids more accurately reflected disease progression in vivo. In addition to hypoxia and ISO, we also included TGFβ$_1$ as excess signaling through this pathway is known to play an important role in the progression of heart failure[28] (Fig. 4A).

Treatment with ISO/hypoxia/TGFβ$_1$ [cardiac injury (CI) stimuli] induced an upregulation of the expression of glycolysis-related genes *GAPDH, LDHA*, and *GLUT1*, and a reduction in expression of *CPT1B*, a mitochondrial transporter of fatty acids within the cardiomyocyte population (Fig. 4B). These changes suggest that the cells are decreasing their capacity for fatty acid oxidation and increasing anaerobic glycolysis, recapitulating the metabolic shift observed in many types of heart disease.

The treated cardiomyocytes also showed elevated levels of *BNP*, a clinical marker of heart failure, a reduction in CX43 expression and downregulation of several calcium handling and ion channel related genes, changes commonly observed in the failing heart[29] (Fig. 4B, C, Supplementary Fig. 4A, B). Notably, the expression level of BNP at baseline in VCM-aggs was relatively higher than that in cardiomyocytes within the organoids The elevation of BNP by the injury stimuli was not observed in the VCM-aggs (Supplementary Fig. 4C). To further validate the 'injured' phenotype of the treated organoids, we next measured CK release, an assay used to detect muscle injury. The outcome of these analyses showed that mature organoids exposed to the CI stimuli released significantly higher levels of CK than the non-treated mature organoids or the immature organoids. The VCM-aggs did not respond to the CI treatment (Fig. 4D). Seahorse analyses showed that the mature organoids treated with the CI stimuli had significantly reduced maximal and spare OCR capacity compared to the untreated organoids indicating that their capacity for oxidative respiration is reduced, a change observed in cardiomyocytes in the failing heart (Fig. 4E). No differences were detected in the treated and non-treated immature organoids. Collectively, these findings show that cells in the mature organoids most accurately recapitulate these key changes associated with cardiovascular disease.

Treatment with the CI stimuli also had a profound effect on the CD90$^+$ fibroblasts as they showed increased expression levels of many ECM genes including *COL1A1, COL3A1, ELN* and *COL5A3* (Fig. 4F). Many of these genes have been found to be upregulated in the failing heart. The expression levels of genes indicative of activated fibroblasts (myofibroblasts) such as *TNC, ACTA2, PALLD*, and *CNN1* as well as those associated with transition to the matrifibrocyte state including *CHAD, COMP*, and *CILP2*, were also upregulated in these fibroblasts[30]. Finally, the expression level of *LOX*, the gene encoding lysyl oxidase, an enzyme inversely correlated with LV ejection fraction in patients with heart failure, was also increased in the activated fibroblasts[31]. Changes in expression of the majority of these fibrosis-related genes were not observed in CI-treated fibroblasts generated directly from the epi cells (Supplementary Fig. 4D, E). Immunohistochemical analyses confirmed the gene expression changes in the organoids and showed increased deposition of FN, COL1, and COL3 in the treated organoid compared to the untreated ones (Fig. 4G, Supplementary Fig. 4F, G). These images clearly demonstrate that the increased FN1 deposition is co-localized with the GFP$^+$ population, suggesting that fibroblasts are the main source of ECM production. Taken together, these findings provide evidence that the cardiomyocyte and fibroblast populations within the

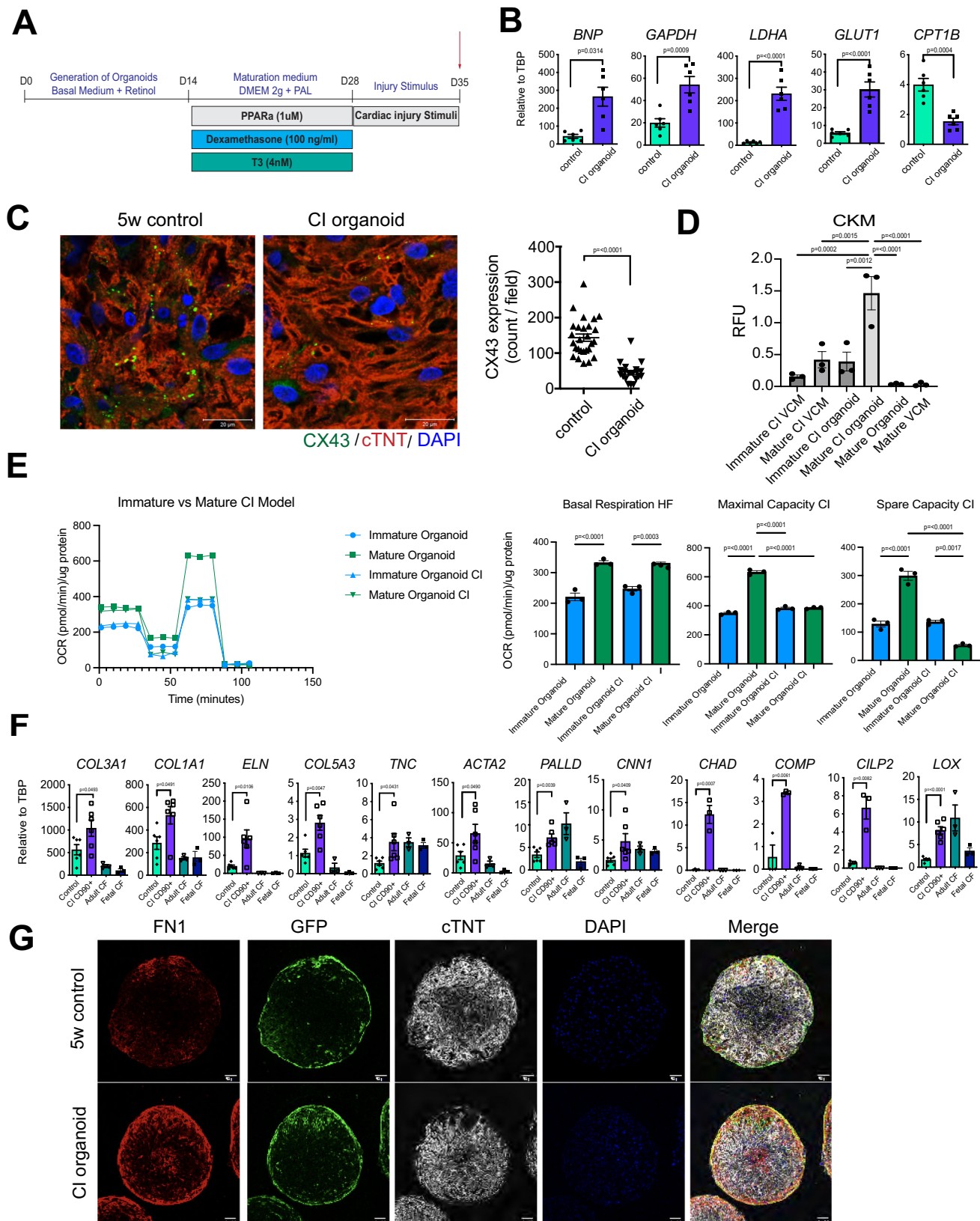

organoids respond to injury stimuli and faithfully recapitulate changes associated with cardiac injury.

## Single-cell RNA sequencing analyses of the cardiac organoids

To assess the cellular heterogeneity of our organoid populations we performed single-cell RNA sequencing (scRNAseq) on control and cardiac injury (CI) organoids at 5 weeks of culture. To track the RFP labeled cardiomyocyte and GFP labeled epicardial-derived stromal populations, we used the lipid-based multiplexing strategy MULTI-seq, as previously described[32] (Supplementary Fig. 5A). With this approach, we showed that, as expected, most stromal (cTNT-negative) cells were GFP-positive whereas the cardiomyocytes (cTNT-positive) were RFP-

**Fig. 4 | Modeling cardiac injury in cardiac organoids. A** Protocol to induce cardiac injury in the cardiac organoids. **B** RT-qPCR expression analyses of the heart failure marker (*BNP*) and metabolic-related genes (*GAPDH, LDHA, GLUT1, CPT1B*) in the cardiomyocyte populations isolated from the control (non-treated) and CI organoids (*N* = 6 biologically independent samples). **C** Left; Representative immunostaining of CX43 in the control and CI-treated organoids. Scale bar; 20um. Right; Quantification of CX43 expression in the two organoid populations (*N* = 27 from 3 independent biological replicates). **D** Quantification of CK release from the indicated CI-treated and control populations (*N* = 3 biologically independent samples). **E** Analyses of the oxygen consumption rate (OCR) in the indicated populations using the Seahorse fatty acid oxidation (FAO) assay (*N* = 3 biologically independent samples each). **F** RT-qPCR expression analyses of extracellular matrix (*COL1A1,*

*COL3A1, ELN, COL5A3*), cytoskeleton (*ACTA2, PALLD, CNN1*), fibrosis-related (*TNC, LOX*), and matrifibrocyte-related (*CHAD, COMP, CILP2*) genes in the FACS isolated CD90⁺ cardiac fibroblast populations from the control and CI organoids (*N* = 3 biologically independent samples for matrifibrocyte genes, *N* = 6 biologically independent samples for others). Adult and fetal CF RNA was included as a reference (*N* = 3 each). **G** Representative immunostaining of fibronectin (FN) in the control and CI organoids at 5 weeks of culture. Scale bar; 20 μm. (*N* = 5 biologically independent samples). Statistical analysis was performed by one-way ANOVA with Tukey's multiple comparisons in (**D**), (**E**) and by two-sided unpaired *t*-test in (**B**), (**C**), (**F**). All error bars represent SEM. CF cardiac fibroblasts, CI organoid cardiac injury organoid, VCM ventricular cardiomyocytes, CKM Creatine Kinase Muscle. Source data are provided as a Source Data file.

positive (Supplementary Fig. 5B). The RFP-positive population from the cardiomyocyte fraction contained some contaminating fibroblasts, which were excluded from downstream analyses. Clustering analysis identified seven cell types and two proliferating populations (Fig. 5A, B, Supplementary Data 1). Analyses of the epicardial-derived lineages revealed that both the control and CI organoids contained clusters with expression profiles indicative of pericytes/smooth muscle cells, including elevated levels of *ACTN1, CALD1, PDGFRB, RGS5*, and *TPM2* (Supplementary Fig. 5C). The finding that the majority of the EPDC population was THY1 (CD90)-positive fibroblasts corroborated our flow cytometry results. These cells broadly expressed ECM genes, such as *COL1A1, COL3A1, FN1* and *POSTN* indicative of the fibroblast lineage (Fig. 2B, C, Supplementary Fig. 5C). We additionally identified a subpopulation (cluster 6) that expressed several adipocyte markers including *PPARG, KLF5, BMP2, CEBPA, CD24*, and *DGAT2* confirming that the epicardial derivatives in our cardiac organoids contained adipocyte-like cells in addition to fibroblasts and smooth muscle cells, recapitulating the developmental potential of the epicardium in vivo[33] (Supplementary Fig. 5D).

We first focused on cardiomyocytes and identified eight transcriptionally distinct clusters in both control and CI organoids (Fig. 5C, D, Supplementary Data 2). These cardiomyocyte subpopulations were enriched in a variety of distinct pathways, including wound healing, apoptotic process, stress response and angiogenesis (Supplementary Fig. 5E). Clusters 7 and 8 represent proliferating cardiomyocytes marked by the expression of *MKI67* and *TOP2A* (Fig. 5E).

To identify transcriptional changes and pathways involved in the cardiac injury response, we selected differentially expressed genes in each cluster between control and CI organoids and performed pathway enrichment analysis. This analysis uncovered changes common to all clusters as well as those unique to specific subsets of cells. Changes found in all clusters included an upregulation of genes related to glycolysis, NADH regeneration and response to hypoxia and a downregulation of genes associated with t-tubule organization, oxidative phosphorylation, and fatty acid beta-oxidation (Fig. 5F, G). Significant increases in the levels of *NPPA* and *NPPB* expression were also detected in the CI-treated organoids compared to untreated controls (Supplementary Fig. 5F). These findings indicate that the cardiomyocytes in all the clusters recapitulate changes commonly observed in heart failure in vivo including the reactivation of a fetal gene signature. Changes specific to a select subset of clusters included the upregulation of expression of genes involved in angiogenesis in cluster 2, inflammation in cluster 4, autophagy in cluster 5, and apoptotic processes in clusters 3, 4 and 5 (Fig. 5F).

To determine if these responses correlated with the maturation state of the cells within a particular cluster, we developed a metabolic maturation score using the maturation signature identified in our previous study[20] and applied it to the different clusters in the organoid population. This analysis showed there was a correlation with maturation as the cells in clusters 1, 3, 4, and 5 with the above expression patterns contained the most metabolically mature cardiomyocytes (Fig. 5H). Integrated analyses between the profiles of the

control and CI-treated organoid and the single-cell RNA seq data on cardiomyocytes from healthy and coronary heart failure patients reported by Wang et al. showed overlap on the UMAP embedding suggesting transcriptional similarities are present (Supplementary Fig. 5G)[33].

To further compare the molecular changes observed in the CI organoids to those found in cardiomyocytes isolated from failing hearts, we generated an in vivo human heart failure signature based on the data set from the study of Wang et al. [34] and scored cardiomyocytes in each of 6 clusters in the organoids. As shown in Fig. 5I, the clusters containing the most mature cells (cluster 1, 3, 4, and 5) in the CI-treated organoids showed the highest degree of transcriptional similarity to the heart failure cardiomyocytes. Additionally, the clusters in the CI organoids were more comparable to the primary heart failure cardiomyocytes than those in the control non-treated organoids. These data suggest that responses of the cardiomyocytes to the pathological stimuli in the organoids recapitulate, to some degree, changes associated with the failing adult heart. Collectively, these findings show that it is possible to model key molecular changes associated with heart failure in cardiac organoids in vitro and that these changes are most accurately recapitulated in metabolically mature cardiomyocytes.

## Characterization of cardiac fibroblast heterogeneity in cardiac organoids
To establish the organ identity of the fibroblasts generated in our cardiac organoids, we compared the expression profiles of those from the control organoids to the expression profiles of fibroblasts from different human fetal tissues. For these analyses, we used a tissue-specific fibroblast signature score that we developed by using the top 100 DEGs amongst the fibroblasts from each tissue, based on data from the human fetal cell atlas[35] and compared it to the organoid fibroblasts. Organoid fibroblasts were most similar to those in the heart, indicating that they display transcriptional features of cardiac fibroblasts (Supplementary Fig. 6A).

To characterize the transcriptional heterogeneity present within the fibroblast population, we subclustered COL1A1-positive and GFP-positive cells from the control and CI organoids (Fig. 6A, Supplementary Data 3). We identified 12 transcriptional fibroblast states and one proliferating subpopulation (MKI67-positive) in both populations. The CI organoids showed an increase in frequency of clusters 1, 2, 3, 4, and 5 and a reduction in frequency of clusters 6, 12, and 13 (Fig. 6B, C). Pathway enrichment analysis of the clusters in both control and CI populations showed heterogeneous development of fibroblasts with a wide range of biological processes common to both including heart contraction, regulation of apoptotic processes, angiogenesis, and heart development (Supplementary Fig. 6B). To compare the expression patterns of the fibroblasts in the control and CI organoids to those of fibroblasts isolated from normal and failing adult human hearts, we integrated the hPSC-derived subpopulations with their in vivo counterparts. These analyses showed transcriptional similarity between fibroblasts within the organoids and those found in the adult human heart (Supplementary Fig. 6C).

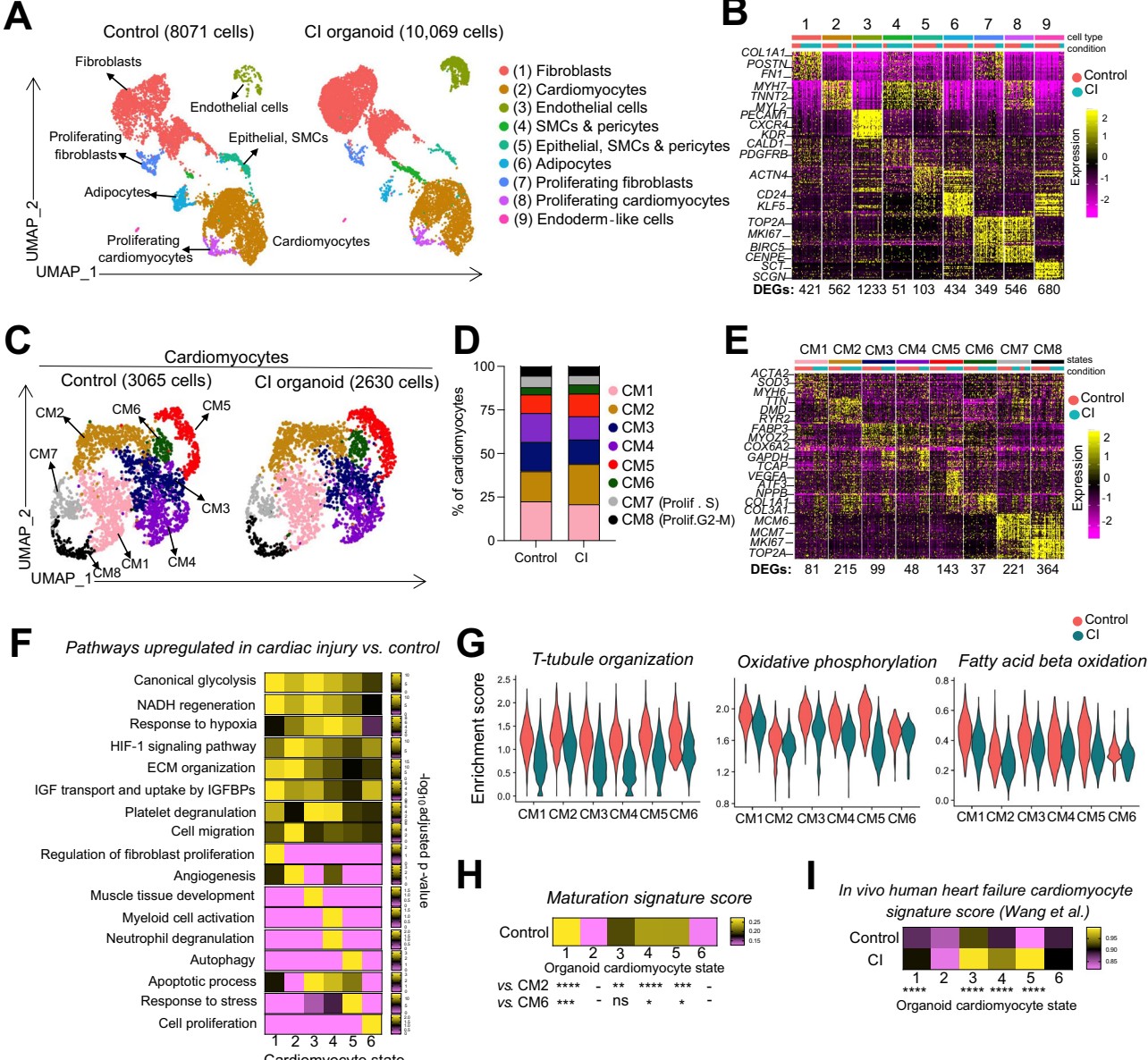

**Fig. 5 | Single-cell RNA sequencing analyses of the cardiac organoids. A** UMAP dimensionality reduction of total cells in control and CI samples. Clusters were annotated based on canonical cell type markers. **B** Heatmap depicting top 30 genes in each cluster in (**A**) (logFC threshold = 0.3, min.pct = 0.3, adjusted *p*-value < 0.05). The number of DEGs in each cluster relative to all other clusters is indicated below the heatmap. Color bars indicate cell type (top) and condition (bottom). **C** Cardiomyocytes were sub-clustered. UMAP dimensionality reduction of cardiomyocytes in control and CI populations. **D** Relative frequency of each cardiomyocyte cluster in control and CI organoid populations. **E** Heatmap depicting top 30 genes in each cardiomyocyte cluster (logFC threshold = 0.3, min.pct = 0.3, adjusted *p*-value < 0.05). The number of DEGs in each cluster relative to all other clusters is indicated below the heatmap. Color bars indicate cluster(top) and condition (bottom). **F** Pathway enrichment analysis (gProfiler, GO: Biological Processes) was performed using DEGs upregulated in cardiac injury vs. control in each cardiomyocyte cluster. Heatmap depicts the enrichment score (-log₁₀ of adjusted *p*-value) scaled across clusters for each pathway individually. **G** Pathway scores were generated using genes annotated for each term in the GO: Biological Processes database. Violin plots show single cell expression of each cumulative pathway score in each cluster for control and CI populations. **H** A cardiomyocyte maturation score was generated using 8 maturation signature genes[20]. Heatmap displays the maturation signature score in each cardiomyocyte cluster in the control sample. One-way ANOVA was performed for each cluster compared to the lowest scoring cluster (CM2 and CM6). **I** An in vivo human heart failure signature was generated using the top 25 DEGs upregulated in ventricular cardiomyocytes in coronary heart failure vs. normal hearts (single cell dataset published in Wang et al.). Each cell in all clusters of control and CI organoids were scored using this signature and displayed in the heatmap. A two-sided unpaired *t*-test was performed for each pairwise comparison (cardiac injury vs. control in each cluster). For differential expression analysis (**B**, **E**, **F**, **H**), Wilcoxon Rank Sum Test (unpaired, two-sided) was performed, and *p*-values were adjusted for multiple comparisons using Bonferroni correction. SMC smooth muscle cells, CM cardiomyocyte, DEG differentially expressed genes. ****p < 0.0001, ***p < 0.001, **p < 0.01, *p < 0.05.

Pathway analyses of differentially expressed genes between the fibroblast clusters in the control and CI organoids revealed an upregulation of genes involved in ECM organization, collagen metabolic processing, supramolecular fiber organization, angiogenesis, and blood vessel development in most of the clusters in the CI population.

Some genes including those associated with wound healing, IGF1-R signaling, response to hypoxia, and cell migration were upregulated in specific clusters, suggesting both common and heterogenous responses to the cardiac injury stimuli in the cardiac fibroblast populations (Fig. 6D). Analyses of genes known to regulate fibrotic

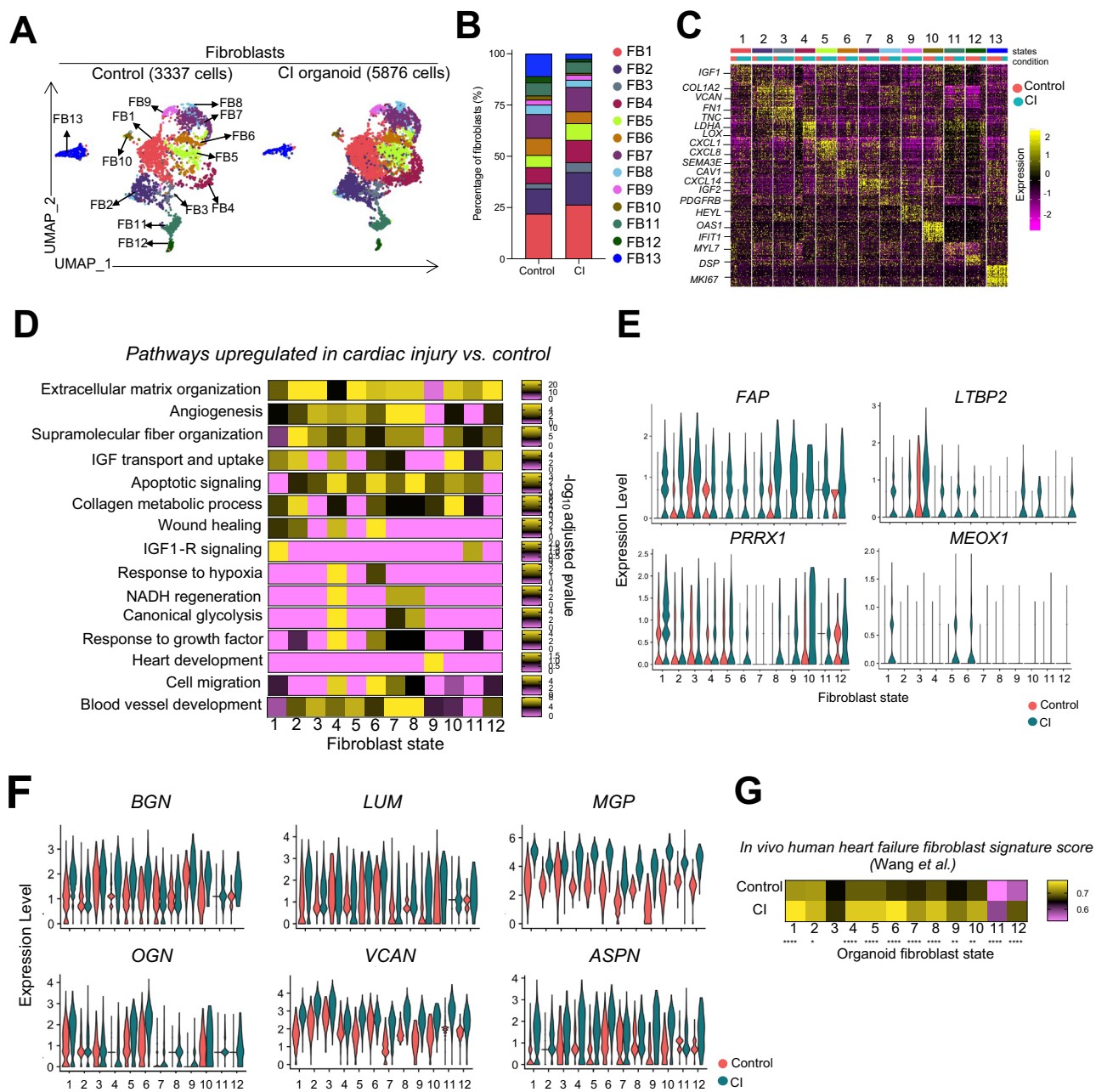

**Fig. 6 | Molecular characterization of fibroblasts in the cardiac organoids.**
**A** UMAP dimensionality reduction of fibroblasts in control and CI samples.
**B** Relative frequency of each fibroblast cluster in control and CI samples.
**C** Heatmap depicting the top 30 genes in each fibroblast cluster (logFC threshold = 0.3, min.pct = 0.3, adjusted p-value < 0.05). Color bars indicate cluster (top) and condition (bottom). Differential expression was performed using the Wilcoxon Rank Sum Test (unpaired, two-sided), and p-values were adjusted for multiple comparisons using Bonferroni correction. **D** Pathway enrichment analysis (gProfiler, GO: Biological Processes) was performed using DEGs upregulated in CI vs. control in each fibroblast cluster. Heatmap depicting the enrichment score ($-\log_{10}$ of adjusted p-value) scaled across clusters for each pathway individually. **E** Violin

plots showing expression of fibrosis-related genes (*FAP, LTBP2, PRRX1, MEOX1*) in each fibroblast cluster in the control and CI samples. **F** Violin plots showing expression of proteoglycan-related genes (*BGN, LUM, MGP, PGN, VCAN, ASPN*) in each fibroblast cluster in the control and CI samples. **G** An in vivo human heart failure signature was generated using the top 25 DEGs upregulated in cardiac fibroblasts in coronary heart failure vs. normal hearts (single cell dataset published in Wang et al.). Each cell in all fibroblast clusters of control and CI organoids was scored using this signature and displayed in the heatmap. A two-sided unpaired t-test was performed for each pairwise comparison (cardiac injury *vs.* control in each cluster). ****p<0.0001, ***p<0.001, **p<0.01, *p<0.05.

responses such as *FAP, LTBP2, PRRX1* and *MEOX1*[10,36], those indicative of fibroblast activation, *COL1A1, COL3A1, ACTA2,* and *CNN1*, and those associated with the matrifibrocyte state including *CILP2* and *COMP* were all expressed at higher levels in the fibroblasts from the CI organoids than in those from the controls indicating that the stimuli used triggered a fibrotic response (Fig. 6E, Supplementary Fig. 6D).

Additionally, we observed elevated levels of expression of various chondroitin sulfate proteoglycans such as *LUM, BGN, MGP, OGN, VCAN* and *ASPN* in the fibroblasts in the CI organoids (Fig. 6F). These changes are in line with findings from a recent study that showed increased levels of these proteoglycans in fibroblasts of the adult human failing heart[37].

We next compared the molecular profiles of both the control and CI organoids to the top DEGs in the fibroblast cluster from patients with ischemic heart failure to determine if molecular changes in the organoid fibroblasts reflected changes associated with heart disease[34]. As shown in Fig. 6G, the profiles from the ischemic heart fibroblasts aligned more closely to the fibroblasts from the CI organoids than to those from the control organoids. Together, these findings demonstrate that the organoid model can recapitulate molecular changes in the fibroblast population associated with ischemic heart disease.

### Identification of a CD9+ reparative like fibroblast in the injured heart

To determine if the CI organoids contain the human equivalent of reparative fibroblasts, we generated a molecular score for the CTHRC+ reparative fibroblasts (RCF score) using the dataset reported by Ruiz-Villalba et al. [15] and applied it to the clusters from the organoids. As shown in Fig. 7A, clusters 1-6 and cluster 10 showed higher RCF scores than the other clusters, suggesting that they may contain reparative fibroblasts. To further investigate this, we next compared the reparative clusters (1–6, 10) and non-reparative clusters (7–9, 11, 12) to a transcriptional profile of WNTx+ reparative fibroblasts characterized by Farbehi et al. [14]. This comparison showed that the putative reparative clusters of fibroblasts from the organoids expressed significantly higher levels of these genes than the non-reparative (canonical) fibroblasts, further supporting the interpretation that the CI organoids contain a subpopulation of reparative fibroblasts (Supplementary Fig. 7A). GO pathway analysis revealed elevated expression levels of genes associated with angiogenesis, blood vessel development, collagen organization and fibroblast proliferation in the reparative populations compared to the canonical fibroblast populations, indicating that these cells display molecular characteristics of the reparative fibroblasts identified in mouse models of heart failure (Fig. 7B).

We next reanalyzed the published mouse data sets for cell surface markers expressed by the reparative fibroblasts. From this analysis, we found that *Cd9* is expressed in both the *Wif1+* and *Cthrc1+* reparative cells (Supplementary Fig. 7B, C). To determine if CD9 is also expressed in fibroblasts in human ischemic failing hearts we analyzed the fibroblasts from the Wang et al. report[34] and found a significant upregulation of CD9 in human ischemic cardiomyopathy fibroblasts compared to those from normal hearts (Fig. 7C).

Flow cytometric analyses showed that both the control and CI organoids contained a subpopulation of CD90+ cells that expressed low levels of CD9. The frequency of CD9+ cells in the CI fibroblast population was approximately double that found in the controls (Fig. 7D). RT-PCR analyses of FACS isolated CD9 populations from the CI organoids showed that the expression levels of genes associated with the subpopulation of reparative fibroblasts in the mouse including *WIF1, DKK3, ATP1B1, DDAH1, FMOD, SOX9* and *COMP* were significantly higher in the CD9+ than in the CD9- cells (Fig. 7E). As expected, the levels of *CD9* were higher in the positive fraction confirming separation based on CD9 expression (Supplementary Fig. 7D). The CD9+ cells also expressed higher levels of genes associated with angiogenesis including *VEGFA, TIMP3, CXCL8, FN1* and *VCAM1*, suggesting a role in supporting vascular development. Differences were also detected in the levels of *IL33*, a cytokine that displays cardioprotective effects on cardiomyocytes and *AGRIN*, an extracellular matrix protein that promotes cardiomyocyte proliferation[9,38,39] (Fig. 7E). Other fibroblast related genes including *WT1, TCF21, TBX18* and *COL1A1* were expressed at comparable levels in the CD9+ and CD9- subpopulations (Supplementary Fig. 7D). Together, these expression patterns support the interpretation that the CD9+ fraction of the CD90+ population in our organoids represents the human equivalent of the reparative fibroblasts described in the mouse and suggest that these cells may have roles in matrix production, angiogenesis, and repair of cardiac tissue (Fig. 7F).

## Discussion

Cardiac fibroblasts play a central role in heart development, in the maintenance of normal heart function throughout adult life and in the progression of different forms of heart disease. Despite their importance in these processes, our understanding of human cardiac fibroblasts, including their interactions with cardiomyocytes, the heterogeneity of the population, the regulation of scar formation and the onset and progression of fibrosis in disease states is limited, largely due to the inaccessibility of human heart tissue. In this study, we have established a simple organoid system with hPSC-derived cells to model key aspects of fibroblast development and maturation and their interactions with cardiomyocytes. With this approach, we were able to document the following characteristics of hPSC-derived fibroblasts that recapitulate known fibroblast behavior and function in the human heart and in the hearts of model organisms. First, we show that the interaction of epicardial cells and cardiomyocytes leads to the specification of fibroblasts that migrate into and seed the organoid tissue where they mature with time in culture or in response to maturation factors. Second, we demonstrate that the fibroblasts within the organoids can respond to pathological stimuli and undergo changes that reflect changes observed in fibroblasts in the failing heart. Third, our single-cell RNA-seq analyses revealed an unappreciated degree of heterogeneity within the fibroblast population and identified a subpopulation that displays molecular profiles of reparative fibroblasts described in the mouse. Together, these findings highlight the power of the hPSC organoid system to accurately model key aspects of heart development and give rise to cell populations that represent those found in both the fetal and adult organ.

Cardiac fibroblasts are generated from epicardial cells through an EMT process initiated by their interaction with the adjacent cardiomyocytes[5]. Here, we modeled this interaction to generate hPSC-derived cardiac fibroblasts, reasoning that this approach would yield a population that most closely represents the fibroblasts found in the human fetal and adult hearts. A key to the success of this strategy was the generation of an epicardial population that was properly patterned and generated from an appropriate mesoderm population. Previous studies on hPSC-derived epicardial cells have used the expression of WT1 as a defining marker of the lineage[40,41]. However, this transcription factor is expressed in different mesothelial populations that derive from different mesoderm subtypes[42]. We monitored Raldh2 (*ALDH1A2*) expression as an additional marker of the epicardial lineage as it is expressed in the fetal epicardium of the mouse, zebrafish and human[43] and the function of this enzyme is essential for the ROH-induced proliferation of the cardiomyocytes. Our demonstration that the addition of ROH at the mesoderm stage of development enhances the generation of RALDH2+ epicardial cells is in line with observations in the mouse and chick that the epicardial progenitors are exposed to retinoid signaling in vivo[44]. The finding that RALDH2+ epicardial cells are preferentially generated from mesoderm that gives rise to atrial cardiomyocytes provides a rationale for optimizing early induction stages for generating cells of this lineage. With access to this optimized population, we were able to recapitulate many aspects of epicardial development and function in the organoid model including the segregation of the cells to the periphery of the tissue, the promotion of proliferation of the adjacent cardiomyocytes and the capacity to undergo TGFβ mediated EMT to generate the spectrum of EPDCs found in the adult heart including fibroblasts, smooth muscle cells and adipocytes. Several recent studies have investigated hPSC-derived epicardial cell-cardiomyocyte interactions in vitro either in monolayer or 3D organoid culture formats[45,46]. Consistent with our observations, the findings from these studies showed that the epicardial cells can promote proliferation of the cardiomyocytes and can undergo EMT to generate derivative mesenchymal cells. However, these studies did not use well characterized populations, nor did they investigate in detail, the derivative lineages generated in the cultures.

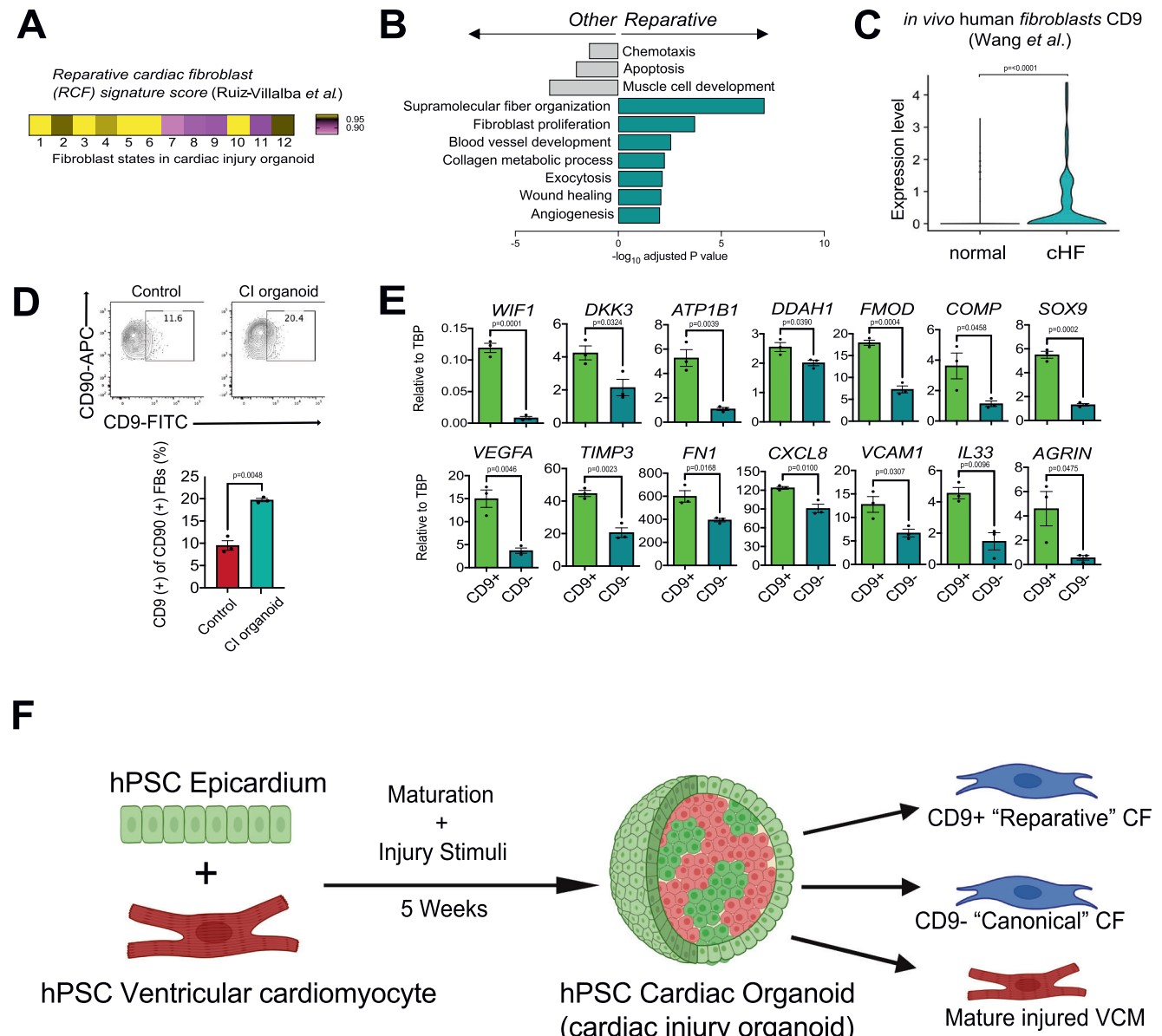

**Fig. 7 | Identification of a CD9⁺ reparative like fibroblast population. A** A reparative cardiac fibroblast signature was generated using the top 25 DEGs of a reparative cardiac fibroblast cluster (as defined by Ruiz-Villalba et al.) vs. other fibroblasts in Ruiz-Villalba et al. Each cardiac fibroblast cluster from the CI population was scored with this signature and displayed in the heatmap. **B** Fibroblast clusters were assigned as "reparative" or "other" based on their relative enrichment of the reparative signature in (**A**). DEGs were computed between these two groups in the CI organoids and pathway enrichment analysis was performed (gProfiler, GO: Biological Processes). The enrichment score (−log₁₀ of adjusted p-value) of the top pathways in each group are displayed. **C** Violin plots showing expression of *CD9* in human cardiac fibroblasts in healthy hearts and ischemic failing hearts (coronary HF (cHF)) (single cell dataset published in Wang et al.). **D** Upper; Representative

flow cytometry analyses of CD9 expression in the CD90⁺ cardiac fibroblast population in the control and CI organoids. Lower; Quantification of the proportion of CD9⁺ cells in the CD90⁺ cardiac fibroblast population in the control and CI organoids (N = 3 biologically independent samples). RT-qPCR expression analyses of indicated genes in the CD9⁺ and CD9⁻ cardiac fibroblast populations isolated (FACS) from the CI organoids (N = 3 biologically independent samples). **F** Schematic summary of cardiac organoid modeling detailing the heterogeneity of canonical and reparative like fibroblasts observed in the cardiac injury conditions. CF cardiac fibroblast, VCM ventricular cardiomyocyte, CI organoid cardiac injury organoid. Statistical analysis was performed by two-sided unpaired t-test in (**C**)–(**E**). All error bars represent SEM. Source data are provided as a Source Data file. Created with BioRender.com.

The immature status of hPSC-derived cardiomyocytes has been recognized as a major hurdle in their use to model different forms of heart disease and to develop new cell-based therapies to treat them[47]. To address this issue, different strategies including electro-mechanical stimulation and factor-induced metabolic transitions have been developed to promote the maturation changes known to occur in the cardiomyocyte populations in vivo[20,48]. In addition to the cardiomyocytes, there is evidence that the cardiac fibroblasts also undergo maturational changes, including changes in composition of the ECM

produced and changes in the expression of genes encoding different transcription factors and signaling molecules[26]. Our findings show that the fibroblasts in the organoids do mature and recapitulate these changes. Notably, some of the changes including the expression levels of specific transcription factors (*TCF21, GATA4* and *MKI67*) and *NOTCH* appear to be induced by the maturation factors that we previously showed promote maturation of cardiomyocytes[20]. These changes occurred in parallel with the expected changes in the cardiomyocyte population, demonstrating that the fibroblasts respond to the same

cues that regulate metabolic maturation of cardiomyocytes. The expression levels of other genes, in particular those that encode matrix proteins did not change in response to these factors, but rather changed with time in culture. The pathways regulating these changes are currently not known. Access to organoids that contain matured cardiomyocytes and fibroblasts enables one to model disease processes that approximate those of the adult heart. Indeed, we were able to demonstrate that treatment of the organoids with pathological stimuli induced changes in both the cardiomyocyte and fibroblast lineages associated with cardiac injury in vivo. Some of these parameters, including the release of CKM and the reduction in oxidative capacity were only observed in the mature organoids, highlighting the importance of using the appropriate cell types and culture format to model disease. Our single cell analyses revealed a high degree of heterogeneity within the fibroblast and cardiomyocyte populations in both the control and CI organoids, recapitulating the heterogeneity of these populations documented in human hearts and in hearts of model organisms[14,15,49]. Our comprehensive analyses of each of the lineages provides important insights into the composition of the populations and their response to pathological stimuli. Notable amongst these was the observation that the organoid-generated cardiomyocytes and fibroblasts showed similar molecular profiles to corresponding cells from the adult heart, suggesting that the strategy of generating cells by recapitulating early developmental events combined with appropriate maturation conditions yields populations appropriate for disease modeling. This notion is supported by the observation that the most mature cardiomyocytes from the CI but not from the control organoids showed striking transcriptional similarities to cardiomyocytes from a failing heart. Additional evidence that this model captures relevant disease-associated changes is the finding that subclusters of fibroblasts from the CI organoids align molecularly with fibroblasts from a failing heart. The identification of a CD9+ fibroblast population with a reparative molecular signature and the increase in the size of the population in the CI organoids suggest that the function of these cells is conserved between species and that these cells may play some roles in the disease process in humans. Our finding that *Cd9* is expressed in the *Wif1+* and *Cthrc1+* reparative cells in the mouse further validates it as a marker of this subpopulation of fibroblasts and the observation that its expression is upregulated in fibroblasts from the human failing heart adds support to the interpretation that these fibroblasts are involved in heart disease. Reparative fibroblasts were first identified and characterized in the mouse based on the expression of genes that encode factors thought to be important in the repair/regenerative processes in the heart including those that regulate WNT signaling and angiogenesis as well as those reported to be cardioprotective (*IL-33*) or capable of promoting proliferation (*AGRIN*)[9,14,15,38,39]. The expression of genes involved in angiogenesis (*VEGFA, FN1*), and those that control the fibrotic response (*WIF1, DKK3, FMOD*) are overlapping with those of fibroblasts in lower organisms that participate in the regenerative process of the damaged heart[50]. While proposed to be reparative, little is known about the actual function of these cells in the mouse or human as methods to isolate them have not been available to date. Our demonstration that this subpopulation of fibroblasts expresses CD9 and can be isolated by FACS provides the first opportunity to access them and study their function in culture models of cardiac development and animal models of heart disease.

In summary, through appropriate modeling of early developmental events, we have established a hPSC-based organoid model that recapitulates cardiac fibroblast development and maturation. Access to distinct subpopulations of fibroblasts, including CD9+ reparative cells, in both the control and cardiac injury populations provides an unprecedented opportunity to investigate the role of these different cell types in normal cardiovascular development as well as in heart disease. The findings from such studies have the potential to identify subpopulations of fibroblasts beneficial to heart function as well as those that promote disease progression. The identification of such subpopulations will provide a novel platform for developing new approaches and ultimately new therapeutics to target specific cell types to manipulate fibrotic responses in heart disease. Additionally, the ability to isolate and co-transplant fibroblast subpopulations such as the CD9+ reparative cells with cardiomyocytes in infarcted hearts may uncover regenerative properties that promote remuscularization of scar tissue.

## Methods
Our research complies with all relevant ethical regulations.

### Directed differentiation of hPSCs
For ventricular differentiation, we used a modified version of our embryoid body (EB)-based protocol. hPSC populations (HES2, HES2-GFP, HES2-RFP) were dissociated into single cells (TrypLE, Thermo-Fisher) and re-aggregated to form EBs in StemPro-34 media (ThermoFisher) containing penicillin/streptomycin (1%, ThermoFisher), L-glutamine (2 mM, ThermoFisher), transferrin (150 mg/ml, ROCHE), ascorbic acid (50 mg/ml, Sigma), and monothioglycerol (50 mg/ml, Sigma), ROCK inhibitor Y-27632 (10 µM, TOCRIS) and rhBMP4 (1 ng/ml, R&D) for 24 h on an orbital shaker (70 rpm). On day 1, the EBs were transferred to mesoderm induction media consisting of StemPro-34 with the above supplements (-ROCK inhibitor Y-27632) and rhBMP4 (8 ng/ml), rhActivinA (12 ng/ml, R&D) and rhbFGF (5 ng/ml, R&D). At day 3, the EBs were harvested, dissociated into single cells (TrypLE), and re-aggregated in cardiac mesoderm specification media consisting of StemPro-34, the Wnt inhibitor IWP2 (1 µM, TOCRIS) and rhVEGF (10 ng/mL, R&D). At day 5, the EBs were transferred to StemPro-34 with rhVEGF (5 ng/ml) for another 5 days and then to StemPro34 for another 5 days. Cultures were incubated in a low oxygen environment (5% $CO_2$, 5% $O_2$, 90% N2) for the first 10 days and a normoxic environment (5% $CO_2$, 20% $O_2$) for the following 5 days.

For cardiomyocyte only maturation cultures, from day 16 to day 18, the EBs were transferred to DMEM high glucose media with XAV (4 µM, TOCRIS) and then transferred to maturation media (DMEM containing low glucose (2 g/L) with Palmitic acid (200 µM, Sigma), Dexamethasone (100 ng/ml, Bioshop), T3 hormone (4 nM, Sigma) and GW7647 (PPARA agonist, 1 µM, Sigma) for the following nine days. Finally, the EBs were cultured in DMEM containing low glucose supplemented with Palmitic acid (200 µM) alone for the following five days (a total of 32 days). Cultures were incubated in a low oxygen environment (5% $CO_2$, 5% $O_2$, 90% $N_2$) for the first ten days and a normoxic environment (5% $CO_2$, 20% $O_2$) for the following 22 days. From day 10 to day 32, the EBs were cultured in polyheme-coated low binding 10 cm culture dishes on an orbital shaker (70 rpm).

For epicardial differentiation, we used different concentrations of rhBMP4 (3 ng/ml) and rhActivinA (1 ng/ml) from day 1 to day 4, followed by Retinol (2 µM, Sigma), rhBMP4 (10 ng/ml), SB431542 (6 µM, Sigma), CHIR (1 µM, Tocris) until day 6. From day 6 to 10 cells are cultured in StemPro34 medium. On day 10 cells are dissociated into single cells (ColB for 1 h 37 °C followed by TrypLE for 5 min at 37 °C) and replated at a density of 250 k/mL on 10 cm culture dishes in StemPro34 with SB431542 (6 µM, Sigma) for 5 days.

For cardiac fibroblast differentiation day 15 epicardial cells were dissociated into single cells (ColB for 1 h 37 °C followed by TrypLE for 5 min at 37 °C) and replated at a density of 500 k/mL in 12 well plates for 3 days in bFGF (30 ng/mL), TGFB1 (1 ng/mL), EGF (5 ng/mL), and CHIR (1 µM, Tocris) in Fibroblast Medium (25%v/v StemPro34 + 75% v/v IMDM + ITSX 1 µL/mL). At 3 days, the medium was replaced with Fibroblast Medium containing bFGF (30 ng/mL) for an additional 3 days in culture. For maturation experiments or CI injury experiments the cells were grown in maturation medium containing PPDT or the CI injury stimuli cocktail.

## Generation of cardiac organoids

Day 15 hPSC epicardial cells and day 15 hPSC cardiomyocytes were dissociated into single cells and then mixed in a 1:1 ratio of 20,000 cells in total and spun down in a 96-well polyheme coated round bottom plate for 1 min at 500 rpm to form organoids in the well. These organoids were cultured for two weeks in StemPro34 medium containing penicillin/streptomycin (1%, ThermoFisher), ʟ-glutamine (2 mM, ThermoFisher), transferrin (150 mg/ml, ROCHE), ascorbic acid (50 mg/ml, Sigma), monothioglycerol (50 mg/ml, Sigma) and retinol (2 μM, Sigma).

At 2 weeks the organoids were transferred to 10 cm polyheme coated dishes and cultured in DMEM high glucose (4.5 g/L, ThermoFisher) media containing penicillin/streptomycin (1%, ThermoFisher), ʟ-glutamine (2 mM, ThermoFisher), transferrin (150 mg/mL, ROCHE), ascorbic acid (50 mg/ml, Sigma), monothioglycerol (50 mg/ml, Sigma) for 14 days to maintain the cells in an immature state. To promote maturation, the organoids were cultured in DMEM low glucose (2 g/L) media with Palmitic acid (200 μM, Sigma), Dexamethasone (100 ng/ml, Bioshop), T3 hormone (4 nM, Sigma) and GW7647 (PPARA agonist, 1 μM, Sigma)) for 10 days and then in DMEM containing low glucose (2 g/L) with Palmitate (200 μM) alone for an additional 4 days.

## Flow cytometry

The EBs were dissociated by incubation in Collagenase type 2 (0.5 mg/mL, Worthington) in HANKs buffer overnight at room temperature followed by TrypLE treatment for 5 min at 37 °C. Cells were stained for 30 min at 4 °C in FACS buffer consisting of PBS with 5% fetal calf serum (FCS) (Wisent) and 0.02% sodium azide. The following antibodies were used for staining: anti-SIRPa-PeCy7 (Biolegend, 1:1000), anti-CD235a/b-APC (1:100), anti-CD9 (Abcam 1:100), and anti-CD90-APC (BD PharMingen, 1:1000). Aldefluor staining was performed as according to manufacturer's intructions (Stemcell Technologies)[17]. Stained cells were analyzed or sorted using an LSR II Flow cytometer (BD PharMingen) or Aria III CFI cell sorter (BD Biosciences). Data were analyzed using FACS DIVA software (BD) and FlowJo software (Tree Star).

## Immunohistochemistry

The EBs were dissociated as described above and the cells were plated onto 24 well culture dishes pre-coated with matrigel (25% v/v, BD PharMingen). Cells were cultured for 2–3 days and then fixed with 4% PFA in PBS for 15 min at room temperature. Cells were permeabilized and blocked with PBS containing 5% donkey serum, 0.1% TritonX. The following antibodies were used for staining: mouse anti-cardiac isoform of cTNT (ThermoFisher Scientific, 1:200), anti-WT1 (Abcam 1:100), anti-GFP (Rockland 1:200), anti-Ki67 (DAKO 1:200), anti-CX43 (Abcam 1:200), anti-FN (Abcam 1:200), anti-COL1 (Abcam 1:200), anti-COL3 (Abcam 1:200), and anti-MYH11 (Abcam 1:100). For detecting unconjugated primary antibodies, the following secondary antibodies were used: donkey anti-mouse IgG-Alexa488 (ThermoFisher, 1:500), donkey anti-rabbit IgG-Alexa555 (ThermoFisher, 1:500), donkey anti-rabbit IgG-Alexa488 (ThermoFisher, 1:500), donkey anti-mouse IgG-Alexa555 (ThermoFisher, 1:500), donkey anti-mouse IgG-Alexa647 (ThermoFisher, 1:500). Cells were stained with primary antibodies in staining buffer consisting of PBS with 0.1% TritonX, and 5% donkey serum overnight at 4 °C. The stained cells were washed with PBS. The cells were then stained with secondary antibodies in PBS containing 0.1% BSA for 1 h at room temperature followed by DAPI staining. For paraffin sections, tissues were fixed by 4% PFA and embedded. After the deparaffinization and rehydration, heat-induced epitope retrieval was performed followed by immunostaining. Sarcomere length was measured in cTNT+ cardiomyocytes randomly selected from 5 to 10 areas and averaged for each cardiac organoid. The average measurement was obtained by using the distances of 4 consecutive Z lines as shown in Supplementary Fig. 3B using Image J. CX43 expression was measured by counting the number of CX43+ cells in one field of view

(×40 magnification) randomly selected from 5 to 10 areas for each cardiac organoid. FN, COL1 and COL3 expression was measured in each cardiac organoid and normalized by the area (/mm²). Stained cells were analyzed using an EVOS Microscope (ThermoFisher), a Zeiss LSM700 confocal microscope (Zeiss), and Image J software (NIH).

## Quantitative real-time PCR

Total RNA from samples was isolated using RNAqueous-micro Kit including RNase-free DNase treatment (Invitrogen). Isolated RNA was reverse transcribed into cDNA using oligo (dT) primers and random hexamers and iscript Reverse Transcriptase (ThermoFisher). qRT-PCR was performed on an EP Real- Plex MasterCycler (Eppendorf) using a QuantiFast SYBR Green PCR kit (QIAGEN). The copy number of each gene relative to the house keeping gene TBP is shown. Primer sequences are listed in Supplementary Data 4. Commercial adult fibroblasts,(cFib), fetal fibroblasts (cFib) (Lonza) and skin BJ fibroblast (Lonza) were included as a reference.

## CKM release assay

Prespecified treatment groups were collected, and cell lysates were obtained using RIPA buffer, collecting both the lysate and culture supernatant. This samples were processed following manufacturer instructions (ABCAM). CKM phosphorylates creatine to yield creatine phosphate, which dissociates into inosine and inorganic phosphate. In this assay inorganic phosphate reacts with ammonium molybdate to produce phosphomolybdic acid, which is reduced to molybdenum blue. The change in absorbance at 660 nm was monitored with a spectrophotometer (SpectraMAX) and CK activity was expressed as RFU/ug of protein.

## Seahorse XF assay

For the Seahorse XF Mitostress test, several organoids or CM aggregates were plated onto an XFe24 cell culture microplate coated with Cell-Tak at 22.6 μg/ml 24 h prior to the assay. 12 h prior to the assay, we replaced the culture media with substrate-limited media: DMEM 2 g G with 1% B27 (-INS), 1% antibiotics, 1% Glutamine, 1% ʟ-Carnitine. Forty-five minutes prior to the assay, the cells were washed twice with KHB (pH 7.4), and 375 μl/well of Seahorse medium (Modified KHB) was added to the cells. The plate was incubated for 45 min at 37 °C. The assay cartridge was loaded with XF Cell Mito Stress Test compounds (3 μM oligomycin, 4 μM FCCP, 0.5 μM rotenone/0.5 μM antimycin A) 30 min prior to the assay. Following calibration, the XF Cell Culture Microplate was immediately inserted into the Seahorse XFe Analyzer and the XF Cell Mito Stress Test was run. After the measurement of OCR, EBs were lysed using RIPA buffer and OCR was normalized to protein content of the corresponding sample lysate using the Bradford assay according to manufacturer's instructions (ThermoFisher). Data were analyzed using Wave software (Agilent).

## Single-cell RNA sequencing

scRNAseq data generated in this study have been deposited at the GEO database under accession code: GSE221500. https://www.ncbi.nlm.nih.gov/geo/query/acc.cgi?acc=GSE221500.

**Library preparation and sequencing.** Samples were prepared as outlined in the Chromium Single Cell 3′ Reagents Kits User Guide (v3 Chemistry) and loaded onto the v3 10x Chromium system. For multi-seq labeling, cells were trypsinized, barcoded with LMOs and pooled before droplet microfluidic-emulsion with the 10x Genomics Chromium system. Sequencing libraries were generated and processed as described by 10x Genomics. Sequencing data (HiSeq2500) was pre-processed using Cell Ranger to create expression matrices. Raw base call (BCL) files were demultiplexed into FASTQ files, and reads were aligned using STAR. Reads were then filtered, followed by barcode and UMI counting to generate the feature-barcode matrices. In order to

assign cells back to their sample of origin (RFP vs. GFP), count matrices were generated from the lipid-tagged libraries generated using MULTI-seq. Cells were qualified as either singlets, doublets (antibody signal for more than one hashtag), or negative (insufficient antibody signal for either hashtag). Doublets and negative cells were removed from the analysis to avoid ambiguity associated with RFP vs. GFP designation. All singlets proceeded to subsequent analyses.

**Pre-processing and quality control.** The R-based (R 3.6.1) package Seurat v3.1 was utilized for all downstream single cell analyses. Data was filtered by removing genes that were not detected in at least three cells, or cells that expressed less than 200 genes. Candidate doublets or multiplets were excluded by removing cells that expressed >6000 genes. Putative dead or lysed cells were excluded by removing cells in which >40% of transcripts mapped to mitochondrial genes. This permissive cutoff was chosen due to the high level of mitochondrial genes in cardiomyocytes. To address the possibility that dissociation of organoids affected gene expression levels, we removed putative stressed or dying cells which expressed a high percentage of transcripts mapping to dissociation-associated genes (30%). For each of these quality control parameters, we removed outlier cells on the basis of the cell distribution for each measure. Control and cardiac injury datasets were first pre-processed and filtered individually, and then merged for all subsequent analyses using the "merge" function in Seurat.

**Normalization, dimensionality reduction and clustering.** Data was normalized using SCTransform in Seurat v3.1. SCTransform removes technical variation while preserving biological variation, and leverages regularized negative binomial regression. This function serves to normalize the data, select highly variable features, and scale the data. The variance-stabilizing transformation (vst) method in SCTransform was used to select the top 3000 highly variable genes. The percentage of mitochondrial transcripts and the number of counts were selected as parameters for regression. Principal component analysis (PCA) was used for dimensionality reduction. The most statistically significant PCs were selected using an elbow plot displaying the standard deviation of each PC. In order to account for technical variation from batch effects between the control and heart failure samples, we performed integration of the two datasets using Harmony. Here, PCA embeddings were adjusted and utilized in graph-based clustering with the FindNeighbors and FindClusters commands. Finally, Uniform Manifold Approximation and Projection (UMAP) was utilized for non-linear dimensionality reduction and projection to a 2D plot for visualization. To compare the heterogeneity present in our organoids to those in human fetal or human adult heart failure, we integrated our datasets using Harmony to remove batch effects while preserving biological variation.

**Differential gene expression and pathway enrichment analysis.** Clusters were annotated on the basis of differentially expressed genes (DEGs). DEGs were computed with the FindAllMarkers command using the Wilcoxon Rank Sum Test (min.pct: 0.3; logFC threshold: 0.3; adjusted $p$-value < 0.05). The top 30 DEGs (based on logFC) were selected for each cluster and displayed on a heatmap, which was downsampled to a maximum of 50 cells per cluster for visualization purposes. Datasets were annotated based on canonical cell type markers in the human heart. In order to define genes that are induced in cardiac injury relative to control organoids, we first used the "subset" function to isolate each cluster separately. We then used the FindMarkers function to compute differentially expressed genes across conditions (cardiac injury vs. control; min.pct: 0.3; logFC threshold: 0.3; adjusted $p$-value < 0.05). Pathway enrichment analysis was performed either cluster-defining DEGs or DEGs induced in cardiac injury for each cluster. We used gProfiler (https://biit.cs.ut.ee/gprofiler/gost)

to measure over-representation of our DEGs against the Gene Ontology (GO): Biological Processes database (http://www.geneontology.org). We displayed the −log10 of the adjusted $p$-value for each GO term as a measure of the pathway enrichment score.

**Quantification and statistical analysis.** All data are represented as mean ± standard error of mean (SEM). Indicated sample sizes (n) represent biological replicates including independent cell culture replicates and individual tissue samples. For single cell data, sample size represents the number of cells analyzed from at least three independent experiments. No statistical method was used to predetermine the samples size. Statistical significance was determined using a student's $t$ test (unpaired, two-tailed) or one-way ANOVA with Tukey's multiple comparisons in GraphPad Prism 6 software (GraphPad Software). All statistical parameters are reported in the respective figures and figure legends.

### Reporting summary
Further information on research design is available in the Nature Portfolio Reporting Summary linked to this article.

## Data availability
The data supporting the findings of this study are available within the article and its supplementary information files, and from the corresponding author on request. A reporting summary for this article is available as a supplementary information file. Raw scRNA seq data generated in this study has been deposited at the GEO database under accession code: GSE221500. Source data are provided with this paper.

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

## Acknowledgements

We would like to thank past and present members of the Keller lab for their advice and comments on the manuscript and experiments. We would like to thank members of the UHN/SickKids flow cytometry sorting facility, the UHN Pathology Research Program, the Princess Margaret Genomics Center at UHN, and the Advanced Optical Microscopy Facility at the Princess Margaret Cancer Research Tower all in Toronto, Ontario, Canada for their technical expertise. This work was supported by a grant from the Canadian Institutes of Health Research (G.K., FDN159937)

## Author contributions

I.F, S.F. and G.K. conceived the project. I.F. and S.F. performed experiments, analyzed data, and wrote the manuscript. H.H. analyzed the single cell RNA sequencing data. S.E. and G.K. designed the project and wrote the manuscript.

## Competing interests

G.M.K. is a scientific co-founder and paid consultant for BlueRock Therapeutics LP, a paid consultant for VistaGen Therapeutics and a board member of Anagenesis Biotechnologies. All other authors declare no competing interests.
