## [Peer Review File · Nature Communications]

Modeling cardiac fibroblast heterogeneity from human pluripotent stem cell-derived epicardial cellsReviewer #1 (Remarks to the Author):

The study by Fernandes et al reports a new human cardiac organoid system with advanced fibroblast and cardiomyocyte maturation of human pluripotent stem cell derived cardiac cells. These studies build on a previous report from the Keller group on epicardial lineage development from hPSCs and demonstrate heterogeneity of fibroblast populations and injury response. This organoid system has important similarities to previously reported in vivo heterogeneity of cardiac fibroblasts at single cell resolution under different conditions. This cardiac organoid system and injury model also may be useful for future studies of human heart failure using an in vitro system.

Comments.

- 1) Epicardial EMT is described for the early stages of organoid formation (Fig 1-2) but expression of definitive markers of EMT such as Twist and Snai were not shown. Also were there any changes in endothelial or epithelial marker genes?
- 2) Similarly, Tgfbeta is profibrotic and directly induces myofibroblast activation indicated by Acat2 expression. Were there differences in gene expression related to EMT and fibroblast activation seen with ROH versus Tgfbeta treatments that might reflect specific fibroblast subtypes and maturation programs?
- 3) Sarcomeric maturation is not obvious in the figures provided for review and higher magnification images might be helpful. Also, how was sarcomere length determined in Figure 3B? A representative image of these changes in cardiomyocyte maturation might be informative.
- 4) Were any indicators of scar formation seen in the organoid injury model? Specifically, were matrifibrocyte markers reported by Fu et al., JCI 2018 induced? This is worth mentioning in the discussion about the extent to which the organoids model cardiac injury response.
- 5) The supplemental tables in pdf format are not optimal (350 pages). Will searchable excel files be available in the data supplement?
- 6) The GSE# is listed as pending (line 672)

Reviewer #2 (Remarks to the Author):

The paper 'Modeling cardiac fibroblast heterogeneity from human pluripotent stem cell derived epicardial cells' by Fernandes, Funakoshi, and Hamidzada et al. describes an organoid culture platform using human pluripotent stem cell derived epicardial cells (PSC-ECs) to study human cardiac fibroblasts using a developmental biology approach. The results presented provide novel into interactions of cardiac organoid cell populations and disease modeling. The heterogeneity of cardiac fibroblasts is not well-described. There are gaps in establishing fibroblast localization and identity, and also more thorough analysis of the scRNAseq fibroblast populations is needed. This hinders the impact of the data of this study.

Detailed comments on these concerns are as follows:

Comments-

- 1) Fig1 and 2 – Were there other aspects of epicardium-myocardium interactions characterized to establish the model? A key characteristic of these interactions is vasculogenesis. It would be good to see staining quantifications for coronary vessel formation alongside proliferation/total CM number counts (such as Pecam counts).
- 2) Fig 3 – Please provide further analysis of PSC-derived cardiomyocyte maturation to supplement Cx43 junctional maturation. FACS must be performed to show maturational changes in ion channels.

- 3) Fig 3, Supp Fig 3 – Please provide information on troponin and alpha/beta myosin heavy chain levels for sarcomeric maturation. Was there isoform switching indicative of maturation? What was the overall percentage of Troponin positive cells in the organoids?
- 4) Supp Fig 3 – Was there reduction in glycolysis genes? Please also provide metabolic analysis by Seahorse mito stress test to characterize increased oxygen consumption of mature differentiated organoids.
- 5) Fig 4 – Was Col1a1 level also upregulated in mRNA expression?
- 6) Fig 4, Supp Fig 4 – Was increased ECM deposition seen in regions of high fibroblast density? Provide staining characterization of fibroblast cell markers (PDGFRa, etc.) in injury model.
- 7) Overall, a systematic fibroblast population characterization over time in the organoids is needed, similar to data in Figure 1 for Epi and V-CMs. When do the cells begin to express fibroblast markers? Which areas do they localize with or without injury modeling? Immunostaining and FACS analysis are needed.
- 8) Supp Fig 5 – Please show ECM gene expression – Col1a1, Col3a1, FN, etc in fibroblast markers panel.
- 9) Identity of fibroblasts was established by using fibroblast marker gene expression panels from human fetal tissues. Analysis of fibroblast heterogeneity using adult cardiac human tissue gene clusters is required, especially for injury model.
- 10) Fig 6 – What were the key markers used for each of the 13 fibroblast clusters? How do these data compare against published fibroblast heterogeneity data in human adult healthy and diseased hearts?

Reviewer #3 (Remarks to the Author):

Fernandes et al have used organoids comprised of PSC-derived cardiomyocytes and PSC-epicardial fibroblasts to study cellular crosstalk during maturation and response to injury. After optimizing epicardial differentiation through retinoid signaling in cardiac mesoderm, they demonstrate that coculture induces EMT in epicardium, generating mostly fibroblasts and some SMC-like cells, also showing that TGFb antagonism blocks this process. They confirm previous reports that cardiac fibroblasts accelerate the maturation of cardiomyocytes, then go on to demonstrate that a previously derived maturation medium accelerates maturation of both cardiomyocytes and fibroblasts. Using an injury model of hypoxia + catecholamines + TGFb they demonstrate gene expression shifts toward a heart failure signature. Single cell profiling identifies multiple subpopulations of myocytes and fibroblasts with different levels of maturity, again seeing greater similarity to human heart failure in the injury model. Detailed evaluation of fibroblast heterogeneity revealed a small population of CD9+ cells that resembles the mouse reparative fibroblast. They conclude that their organoid system recapitulates key aspects of development and is a new platform to model disease.

General Comments

This paper comes from a good group and contains a substantial amount of interesting data. It is mostly a descriptive study that relies heavily on gene expression patterns. I had trouble finding a logical trajectory/storyline. At the beginning it seemed to be about cardiomyocyte-fibroblast crosstalk, and then it moved into tissue response to injury, finishing in transcriptional cellular subtyping. It would help the reader to offer a clear problem statement or central hypothesis at the beginning, and then organize the paper along that path. As it stands, it is a bit of a meander. Additional points for improvement are listed below.

1. The study focuses on cardiomyocyte-fibroblast organoids. Monotypic control aggregates would

be the appropriate negative controls in this case, but they appear to be used sparingly. Occasional reference is made to cardiomyocyte aggregates, but I did not note any fibroblast aggregate data. Please use monotypic aggregates to support all major conclusions, so that readers understand the advantages that complexity brings.

2. The CD9+ fibroblast appears to be the most significant finding from this investigation. I would like to see a test of its functional significance through gain- and loss-of-function studies. Does it play a reparative role vs. other populations? Does its deletion impair the ability of an organoid to heal after an injury?

3. It would be informative to understand the extent to which electrophysiological remodeling occurs in the organoids in response to coculture, maturation medium, and chemical injury. Please systematically analyse ion channel genes involved in pacemaking, the action potential, and calcium handling in your analyses (recognizing that a few examples are already presented).

4. For studies of tissue injury, it would be important to use standard indices of cell death, beyond gene expression. CK release as a % of total CK would be useful here. Are organoids more or less susceptible to cell death than monotypic aggregates? Does maturation increase or decrease susceptibility to death?

Specific Comments

1. Figure 2. Please present percent-positive cells in the figures in addition to MFI, to agree with the text.

2. Please provide more details on the maturation medium so that readers do not have to review your previous publication.

3. There appears to be only n=1 for the adult cardiac fibroblast studies. This is insufficient for quantitative evaluation.

4. The BNP data are a bit more complex than the text describes. It appears that BNP was downregulated by EPDCs at baseline, and injury brings it up to level of monoculture aggregates. Please explain this behaviour.

Point-by-point responses to the reviewers' comments:

Reviewer #1 (Remarks to the Author):

1. Epicardial EMT is described for the early stages of organoid formation (Fig 1-2) but expression of definitive markers of EMT such as Twist and Snai were not shown. Also were there any changes in endothelial or epithelial marker genes?

Response: We agree with the reviewer's comment that the gene expression patterns of Twist and Snai would be informative to the readers. Findings showing that the expression of both genes is upregulated in the epicardial-derived CD90^{high} fibroblast population is included in Supplementary Figure 2A of the revised manuscript. We have also analyzed expression of CD31 at different timepoints by flow cytometry as a measure of endothelial development. As shown in Supplementary Figure 2B, CD31⁺ cells are not detected at any of the timepoints analyzed strongly suggesting that the day 15 epicardial cells do not undergo differentiation to the endothelial cell lineage in cardiac organoids under the conditions used.

2. Similarly, Tgfbeta is profibrotic and directly induces myofibroblast activation indicated by Acat2 expression. Were there differences in gene expression related to EMT and fibroblast activation seen with ROH versus Tgfbeta treatments that might reflect specific fibroblast subtypes and maturation programs?

Response: We did not directly compare the effects of ROH vs TGF β . We did treat the organoids with ROH and found that it leads to a reduction in the size of the ALDH⁺ population (Figure 2G). Treatment with ROH had no effect on gene expression in the fibroblast population (Supplementary Figure 2I). We have not treated the organoid with exogenous TGF β during the characterization of the model but did show that blocking the pathway with SB431542 does partially inhibit the generation of the CD90^{high} population and maintains the ALDH⁺ epicardial population (Figure 2D, 2E). Blocking the pathway also decreased the expression levels of the fibroblast and smooth muscle genes (Figure 2F). TGF β was used together with isoproterenol and hypoxia as an injury stimulus (Figure 4). Our apologies to the reviewer if this was confusing in the manuscript.

3. Sarcomeric maturation is not obvious in the figures provided for review and higher magnification images might be helpful. Also, how was sarcomere length determined in Figure 3B? A representative image of these changes in cardiomyocyte maturation might be informative.

Response: We agree with the reviewer and have included higher magnification images showing the sarcomere structures measured in Figure 3B. Additionally, we have included a representative image illustrating the methodology used for measuring sarcomere length in Image J in Supplementary Figure 3B.

4. Were any indicators of scar formation seen in the organoid injury model? Specifically, were matrifibrocyte markers reported by Fu et al., JCI 2018 induced? This is worth mentioning in the discussion about the extent to which the organoids model cardiac injury response.

Response: We agree that this is an important point and have analyzed expression patterns of several genes indicative of the transformation of cardiac fibroblasts to a matrifibrocyte state. Our findings show that the levels of expression of the matrifibrocyte markers *CHAD*, *COMP*, and *CILP2*, were all significantly increased in the injured organoids indicating that this model recapitulates the molecular changes associated with the scar formation process. These data are shown in Figure 4F and Supplementary Figure 6D.

5. The supplemental tables in pdf format are not optimal (350 pages). Will searchable excel files be available in the data supplement?

Response: Searchable excel files have been made available in the updated supplementary tables.

6. The GSE# is listed as pending (line 672)

Response: We have received our GSE# and have added this to the main manuscript text.

Reviewer #2 (Remarks to the Author):

1. Fig1 and 2 – Were there other aspects of epicardium-myocardium interactions characterized to establish the model? A key characteristic of these interactions is vasculogenesis. It would be good to see staining quantifications for coronary vessel formation alongside proliferation/total CM number counts (such as Pecam counts).

Response: This a valid point as epicardial to myocardial interactions will induce angiogenesis from the sinus venosus *in vivo* and a small proportion of epicardial cells do indeed trace to coronary endothelium in the developing heart. However, under the culture conditions used, our model does not recapitulate this potential of the epicardial cells as we do not detect any CD31 cells at either days 4 or 14 of organoid culture. These data are shown in Supplementary Figure 2B.

2. Fig 3 – Please provide further analysis of PSC-derived cardiomyocyte maturation to supplement Cx43 junctional maturation. FACS must be performed to show maturational changes in ion channels.

Response: Our original analyses showed changes in the expression levels of sarcomere and metabolism related genes indicative of cardiomyocyte maturation. As there are no validated flow cytometry antibodies available for the quantification of ion channels at the single cell level, we

used RT-qPCR analyses to monitor changes in gene expression as a measure of electrical maturation. The genes analyzed include *KCNK1*, *HCN2*, *KCND3*, *CACNA1C*, *KCNJ2*, *KCNH2*, and *HCN4*. The findings from these analyses, shown in Supplementary Figure 3C, are further supportive of a change in the maturation status of the cardiomyocytes in the organoids

3. Fig 3, Supp Fig 3 – Please provide information on troponin and alpha/beta myosin heavy chain levels for sarcomeric maturation. Was there isoform switching indicative of maturation? What was the overall percentage of Troponin positive cells in the organoids?

Response: We analyzed the expression levels of *MYH7*, *MYH6* and *TNNI1* and *TNNI3* in the immature and mature organoids (Supplementary Figure 3C). The levels of *MYH6* and *TNNI1* were significantly reduced in the mature organoids compare to the immature structures. The levels of *MYH7* and *TNNI3* were elevated in some experiments, but did not reach significance. However, with the changes in *MYH6* and *TNNI1*, we are able to report that the *MYH7/6* ratio and *TNNI3/1* ratio have increased during our maturation process.

The proportion of cardiomyocytes in the organoids did not change significantly following maturation. We have added several flow cytometry plots to the revised manuscript (Supplementary Figure 3A) showing that approximately 50% of the cells are RFP⁺ and of these between 85% and 90% are cTNT⁺MLC2V⁺ cardiomyocytes. The myocyte content of the organoids ranges between 40-45%. Data from our single cell RNA-seq analyses show that the bulk of the RFP⁺ cells remain cTNT⁺ following injury (Figure 5A and Supplementary Figure 5B).

4. Supp Fig 3 – Was there reduction in glycolysis genes? Please also provide metabolic analysis by Seahorse mito stress test to characterize increased oxygen consumption of mature differentiated organoids.

Response: To address this question, we analyzed the expression pattern of several glycolysis related genes, and while there were differences in some experiments, these changes were not statistically significant (Supplementary Figure 3C). We also performed seahorse metabolic functional analysis on the different organoid and VCM aggregate populations. The findings from these analyses showed that the mature organoids displayed the highest oxidative capacity as measured by differences in basal respiration, spare capacity and maximal capacity (Figure 3E).

5. Fig 4 – Was Col1a1 level also upregulated in mRNA expression?

Response: RT-qPCR analyses of the CD90⁺ fibroblast population showed that expression of *COL1A1* was indeed upregulated following injury. These data are shown in Figure 4F of the revised manuscript. Additionally, the scRNAseq data presented in Supplementary Figure 6D shows that that *COL1A1* mRNA expression is increased across the majority of subpopulations of cardiac fibroblasts isolated from the CI-treated organoids.

6. Fig 4, Supp Fig 4 – Was increased ECM deposition seen in regions of high fibroblast density? Provide staining characterization of fibroblast cell markers (PDGFRa, etc.) in injury model.

Response: This is an important point that we have addressed through the comparison of GFP and FN1 expression in the organoids. These expression patterns clearly demonstrate that the regions of elevated FN1 deposition co-localize with the GFP⁺ population (Figure 4G). Our flow cytometry analyses (Figure 2B and 2G) showed that the majority of GFP⁺ cells in immature, untreated organoids at this stage are CD90⁺ fibroblasts. Data from our single cell RNA-seq analyses (Supplementary Figure 5B) indicates that this is also the case for the mature, CI treated structures as most of the GFP⁺ cells were annotated as belonging to the fibroblast lineage. Collectively, these findings support the interpretation that increased ECM deposition is present in regions of elevated fibroblast density. As requested, we did initially attempt to characterize the fibroblast population based on the expression of different markers. For these studies, we evaluated staining patterns with commercially available antibodies for CD90 (Cat#: NB100-65543), CD140a (Cat#: AF1062), and FSP1 (Cat #: NBP2-36431). However, in our hands, none yielded satisfactory signal detection of the targeted epitope. Given this, we chose to monitor GFP expression and feel that the data presented support our interpretation that the majority of GFP⁺ population is fibroblast and the main source of ECM deposition.

7. Overall, a systematic fibroblast population characterization over time in the organoids is needed, similar to data in Figure 1 for Epi and V-CMs. When do the cells begin to express fibroblast markers? Which areas do they localize with or without injury modeling? Immunostaining and FACS analysis are needed.

Response: In our initial analyses, we showed that the epicardial population has initiated the transition to the fibroblast lineage as evidenced by the emergence CD90⁺ALDH⁺ and CD90^{high}ALDH⁻ subpopulations by 4 of organoid culture (Figure 2B, 2C). By day 14, the majority of epicardium has transformed into CD90⁺ALDH⁻ fibroblasts. We have extended these analyses to the 4 week post-maturation and 5 week injured and non-injured organoids and show that the CD90⁺ALDH⁻ fibroblasts remain the predominant population at these times (Supplementary Figure 8A and 8B). Furthermore, we have provided IHC of entire organoids to demonstrate the localization of GFP⁺ fibroblasts with or without injury in Figure 4G, indicating that majority of fibroblasts are located in the outer layer in the organoids while some of fibroblasts are inside the organoids. These findings are similarly observed in the organoids with or without injury.

8. Supp Fig 5 – Please show ECM gene expression – Coll1a1, Col3a1, FN, etc in fibroblast markers panel.

Response: We have added the expression of these genes to Supplementary Figure 5C.

9. Identity of fibroblasts was established by using fibroblast marker gene expression panels from human fetal tissues. Analysis of fibroblast heterogeneity using adult cardiac human tissue gene clusters is required, especially for injury model.

Response: The identity of fibroblasts was assigned after clustering analysis using the canonical markers of the lineage found in Supplemental Figure 5C. We acknowledge the importance of comparing our findings with the characteristics of the adult human heart. To address this, we have added the integration analysis in Supplementary Figure 5G and Supplementary Figure 6C. These analyses indicate that the transcriptional heterogeneity observed in cardiomyocytes and fibroblasts within our organoids is similar to that of the adult human heart.

10. Fig 6 – What were the key markers used for each of the 13 fibroblast clusters? How do these data compare against published fibroblast heterogeneity data in human adult healthy and diseased hearts?

Response: The key markers of each fibroblast cluster are listed in Figure 6C and Supplementary Table 3. Comparison of these data to the primary adult healthy and diseased hearts from coronary artery disease, is included in Supplementary Figure 6C. We have added a more detailed explanation of the methodology pertaining to the integration analysis of our scRNAseq data with the published data in the method section of the main text.

Reviewer #3 (Remarks to the Author):

Major concerns

1. The study focuses on cardiomyocyte-fibroblast organoids. Monotypic control aggregates would be the appropriate negative controls in this case, but they appear to be used sparingly. Occasional reference is made to cardiomyocyte aggregates, but I did not note any fibroblast aggregate data. Please use monotypic aggregates to support all major conclusions, so that readers understand the advantages that complexity brings.

Response: We agree that analyses of monotypic controls is important and have extended our analyses as follows. For the cardiomyocytes, we have compared gene expression patterns in the organoids to the cardiomyocyte aggregates (VCM-agg) following the maturation step and demonstrate comparable changes in both formats (Supplementary Figure 3C). Additionally, we carried out metabolic functional assays (FAO seahorse assay) and show that the mature organoids displayed higher oxidative capacity than the VCM-aggs alone (Figure 3E). Finally, we have measured CK release and demonstrated significantly higher levels of release from the CI-treated organoids than from the treated VCM-aggs (Figure 4D). Together, the findings from these two functional assays support the interpretation that the organoid more accurately recapitulate the disease phenotype than the aggregates of cardiomyocytes.

Analyses of epicardial cell or fibroblast-only aggregates was much more challenging as these cells do not aggregate well and the few small aggregates that did form did not survive well in culture.

To address this question, we generated fibroblasts directly from the epicardial population and analyzed these cells grown in 2D monolayer culture (Supplementary Figure 2C, 2D, and 2E). Unfortunately, even under these conditions the monolayer detached beyond 1 week of maturation culture. They did survive an additional week when switched to the injury stimulus. Analyses of the fibroblasts at these time points revealed that these cells cultured alone showed little response to either the maturation (Supplementary Figure 3E and 3F) or injury cues (Supplementary Figure 4D and 4E), suggesting that these changes are dependent on interactions within the organoids.

2. The CD9+ fibroblast appears to be the most significant finding from this investigation. I would like to see a test of its functional significance through gain- and loss-of-function studies. Does it play a reparative role vs. other populations? Does its deletion impair the ability of an organoid to heal after an injury?

Response: We agree with the reviewer that the identification of the CD9⁺ fibroblast population is of great interest and potentially one of the most significant findings of our study. We also agree that gain- and loss-of-function experiments could provide important insights into the role of this marker in the injury response in the organoids. However, these are extensive, long-term experiments that, in our view, represent the next stage of the project and consequently fall outside the scope of the current study. The findings presented in our revised manuscript are novel and for the first time, describe a system to model human cardiac fibroblast development and to study the heterogeneity of this population following injury. As such, we believe they provide a complete story without the need for additional genetic studies.

3. It would be informative to understand the extent to which electrophysiological remodeling occurs in the organoids in response to coculture, maturation medium, and chemical injury. Please systematically analyse ion channel genes involved in pacemaking, the action potential, and calcium handling in your analyses (recognizing that a few examples are already presented).

Response: We analyzed expression patterns of additional ion-channel gene at the formation of the organoids (week 2) as well as at the maturation (week 4) and injury stages (week 5). These data are included in Supplementary Figures 1H, 3C and 4B respectively.

4. For studies of tissue injury, it would be important to use standard indices of cell death, beyond gene expression. CK release as a % of total CK would be useful here. Are organoids more or less susceptible to cell death than monotypic aggregates? Does maturation increase or decrease susceptibility to death?

Response: We agree that this is an important point and have carried out the CK release assay on the different organoids and aggregates. The findings showed that the mature CI-treated organoids released significantly more CK than the CI-treated immature organoids or either of the VCM-agg groups indicating that they most accurately model the disease phenotype. These data are presented in Figure 4D of the revised manuscript.

Minor concerns

1. Figure 2. Please present percent-positive cells in the figures in addition to MFI, to agree with the text.

Response: We have added these data into Figure 2E and 2G.

2. Please provide more details on the maturation medium so that readers do not have to review your previous publication.

Response: We have added this information to the methods section of the manuscript.

3. There appears to be only n=1 for the adult cardiac fibroblast studies. This is insufficient for quantitative evaluation.

Response: We have included an additional n=2 of the primary cell samples in Figures 2, 3 and 4.

4. The BNP data are a bit more complex than the text describes. It appears that BNP was downregulated by EPDCs at baseline, and injury brings it up to level of monoculture aggregates. Please explain this behaviour.

Response: The baseline levels of BNP are indeed higher in the VCM-aggs than in the organoids at 5 weeks. The reason for this is not entirely clear but may be due to stress on the cardiomyocytes that is reduced by the presence of the non-cardiomyocyte populations, resulting in lower levels of BNP. With this interpretation, the cells in the organoids are able to respond to injury and show increased levels of BNP. The cardiomyocytes in the VCM-aggs, by contrast are already experiencing a stress response and do not show any further increase following the injury.

Reviewer #1 (Remarks to the Author):

The revised manuscript has addressed my concerns and is significantly improved with revisions made in response all the reviewers' comments. I have no additional comments.

Reviewer #2 (Remarks to the Author):

My comments were suitably addressed.

Reviewer #3 (Remarks to the Author):

The manuscript is significantly improved by the revision. I had hoped to learn the significance of the CD9+ fibroblast population, but I accept the authors' statement that this question cannot be answered in a reasonable amount of time. The remainder of my comments were well addressed, and I have no further suggestions for improvement. Congratulations on a nice piece of work.